# Numerical Simulation of Reinforced Concrete Piers after Seawater Freeze–Thaw Cycles

**Fei Teng** [1,†] , **Yueying Zhang** [1], **Weidong Yan** [2,†], **Xiaolei Wang** [2,*] **and Yanfeng Li** [1]

1 School of Transportation and Geometics Engineering, Shenyang Jianzhu University, Shenyang 110168, China
2 School of Civil Engineering, Shenyang Jianzhu University, Shenyang 110168, China
* Correspondence: wangxiaol72@163.com
† These authors contributed equally to this work.

**Abstract:** The reinforced concrete (RC) piers of offshore bridges inevitably experience seawater freeze–thaw cycles due to the periodic movement of tides in cold climates. The damage caused by seawater freeze–thaw cycles will reduce the durability and mechanical properties of concrete, and then affect the seismic performance of RC piers. The method of seismic performance analysis on RC piers by numerical simulation is gradually emerging because the process of the conventional experiment is relatively complicated, and the heterogeneity and degradation of concrete after seawater freeze–thaw cycles should be considered. In this study, the method of meso-element equivalent and layered modeling was used to simulate a low cyclic loading test on an RC pier after seawater freeze–thaw cycles with ABAQUS software. The numerical simulation results were compared with the experimental results; the deviation value of peak load was not more than 6%, and the deviation value of peak displacement was not more than 10%. The result of the numerical simulation matched well with the experimental results, and the influence of different parameters was analyzed through the practical method of numerical simulation. It can be determined that the peak load decreased by 11%, while the peak displacement increased by 40% after 125 seawater freeze–thaw cycles. In the same 125 freeze–thaw cycles, the peak load increased by 15% and 27% while the axial compression ratio and the longitudinal reinforcement diameter increased. As the stirrup spacing of specimens decreased, the peak load remained unchanged, but the ductility coefficient of the specimens increased by 20%.

**Keywords:** seawater; freeze–thaw cycles; low cyclic loading test; numerical simulation; meso-element equivalent

## 1. Introduction

Offshore bridges are located in a marine environment that is more complex than the inland environment, so reinforced concrete (RC) piers suffer from seawater freeze–thaw cycles, dry–wet cycles, and chloride ion corrosion for a very long time during their service period [1,2]. The American Society of Civil Engineers (ASCE) indicated that the freeze–thaw cycle is one of the most critical factors in reducing the durability of concrete materials [3]. The seawater freeze–thaw damage to the concrete of piers mainly occurs in cold regions, such as northern Europe, northern America, Canada, northern Japan, and northern China [4,5]. Although the natural environment varies from region to region, the concrete sustains seawater freeze–thaw damage leading to the deterioration of the durability of RC piers [6–9]. According to the investigation of the offshore bridges along the Bohai Bay Highway in northern China, the pre-installed anti-freezing measures (coating paint, stacking stone brick, and wrapping foam asbestos net) on some RC piers have been damaged in different forms after seawater freeze–thaw cycles, as shown in Figure 1. The concrete of RC piers within the range of tidal action has been damaged to varying degrees after seawater freeze–thaw cycles. The forms of destruction included voids and pits on the

surface of the component, spalling of cover concrete, and corrosion of the reinforcement, as shown in Figure 2.

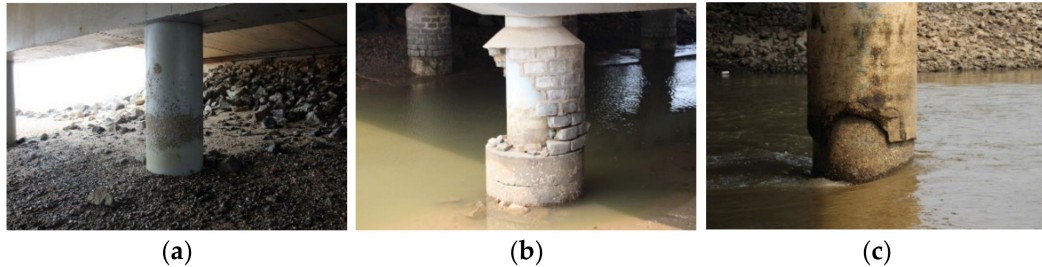

(a)　　　　　　　　　　　　(b)　　　　　　　　　　　　(c)

**Figure 1.** Damage forms of pre-installed anti-freezing measures. (**a**) Coating paint. (**b**) Stacking stone brick. (**c**) Wrapping foam asbestos net.

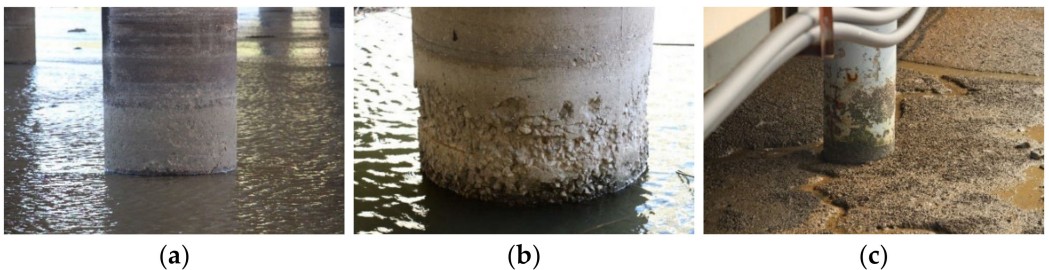

(a)　　　　　　　　　　　　(b)　　　　　　　　　　　　(c)

**Figure 2.** Destruction forms of RC piers after seawater freeze–thaw cycles. (**a**) Voids and pits. (**b**) Spalling. (**c**) Corrosion.

Offshore bridges in cold regions with high seismicity are not only subjected to seawater freeze–thaw cycles but also threatened by intense earthquake action. Earthquake action is highly destructive and could destroy bridges due to releasing a large amount of energy in a short time [10]. Many earthquake damage data show that RC piers are prone to damage under earthquake action as the main component of bridge bearing and lateral force resistance [11]. Taking northeast China as an example, Bohai Bay is located in the Circum-Pacific seismic belt area, which caused the world-famous Haicheng Earthquake in 1975 [12]. The degradation in the durability of concrete would lead to the decrease in mechanical properties on RC piers, which would directly affect the ability of the bridge to resist earthquake action [13]. Therefore, it is necessary to study the seismic performance of RC piers subjected to seawater freeze–thaw cycles in the marine environment to reduce the seismic risk of offshore bridges in the seismic belt area and cold region.

The low cyclic loading test is a standard test method that could be used to analyze the seismic performance of RC piers. Experimental studies of RC piers with different design parameters after freeze–thaw cycles have been carried out. The seismic performance of RC piers after freeze–thaw cycles was studied by Xu et al. [14,15] using a low cyclic loading test, and the failure modes, hysteretic curves, bearing capacity, displacement ductility coefficient, stiffness degradation, and energy dissipation capacity of RC piers under different freeze–thaw cycles and axial compression ratios were compared and analyzed. The freezing and thawing processes of the specimens were completed in the air to simulate the freeze–thaw cycles that occurred in steam and rain environments, and the spray solution was freshwater. The results of experimental studies show that the bearing capacity, displacement ductility coefficient, and energy dissipation capacity of RC piers all decrease to varying degrees with the increase in the number of freshwater freeze–thaw cycles, and the stiffness degradation speed of specimens gradually accelerates. The effect of the axial compression ratio was similar to that of non-freeze–thaw specimens, and the seismic performance of the RC piers decreased with the increase in the number of freshwater freeze–thaw cycles. The scientific research team at Xi'an University of Architecture and Technology took a step

further through a low cyclic loading test based on the above research. Zhang et al. [16] conducted an experimental study on RC piers with different design parameters (concrete strength grade, axial compression ratio, and shear span ratio). An air environment was also chosen for the freeze–thaw conditions. The effects of freshwater freeze–thaw cycles on the seismic performance of RC columns were analyzed by comparing the failure modes, hysteretic cures, flexural capacity, stiffness degradation, and accumulated energy consumption. The results indicate that the failure mode of the RC piers with a small shear span ratio ($\lambda$ = 2.5) gradually changes from flexural failure to shear failure in low cyclic loading tests after freeze–thaw cycles. The experimental results show that the flexural capacity, deformation capacity, and cumulative energy consumption capacity of each specimen deteriorate to different degrees with the increase in the number of freshwater freeze–thaw cycles. Among them, the degradation rate of strength and stiffness gradually accelerated. The flexural capacity and energy dissipation capacity of RC piers increased first and then decreased with the axial compression ratio, the deformation capacity decreased gradually, and the rate of stiffness degradation accelerated. The flexural capacity and energy dissipation capacity of RC piers increased with the concrete strength grade, and the displacement ductility coefficient decreased slightly when freeze–thaw cycles were the same.

A large number of studies show that the preparation process of the specimen is relatively complicated. The period of the rapid freeze–thaw cycle test is relatively long due to the limitation of test conditions and equipment while conducting the experimental research (rapid freeze–thaw cycle test and low cyclic loading test). Numerical simulation has recently been used in the study of the seismic performance of components after freeze–thaw cycles. Liu [17] determined the constitutive relation of concrete material after freshwater freeze–thaw cycles based on the experimental study of the stress–strain relationship of concrete after freshwater freeze–thaw cycles. The ABAQUS software was used to simulate the mechanical behavior of the RC column under a low cyclic loading test after freshwater freeze–thaw cycles. The accuracy of the numerical simulation method was verified by comparing it with the experimental results. Zhang et al. [18–20] conducted a numerical simulation of the shear wall and RC columns after freshwater freeze–thaw cycles using the fiber beam element method. Based on the pull-out test of reinforced concrete specimens, the degradation law of bond strength was established, which could consider the distribution of freeze–thaw damage. The bond-slip calculation method for freeze–thaw damage was verified based on the assumption of simplified bond stress distribution. The fiber beam–column model was proposed, which could consider uneven damage and the bond-slip effect of freeze–thaw damage. The OpenSEES software was used to simulate the low cyclic loading test of the shear wall and RC column after freshwater freeze–thaw cycles, and the results were compared and analyzed to verify the effectiveness of the proposed model and numerical simulation method. The existing research on the seismic performance of piers after freeze–thaw cycles mainly considers the freshwater environment, and the research on the marine environment is limited [18]. Therefore, it is necessary to study the method of the numerical simulation of the seismic resistance of RC piers after seawater freeze–thaw cycles.

The fiber element and finite element methods are generally accepted for the numerical simulation of RC components after freeze–thaw cycles. If we consider the freeze–thaw depth of RC members, the heterogeneity of freeze–thaw damage, and the constitutive relationship of concrete after seawater freeze–thaw cycles in the modeling process, the numerical simulation results could be closer to the experimental results. Based on the existing research, combined with the equivalent meso-element model and layered modeling method, the low cycle loading test of RC piers after seawater freeze–thaw cycles was numerically simulated in this study. The numerical simulation method proposed in this paper reasonably considers the heterogeneity of freeze–thaw damage distribution in concrete and is closer to the damage distribution characteristics of RC piers after seawater freeze–thaw cycles, which could provide a reference for seismic response analysis of offshore bridges in cold regions.

## 2. Constitution of Concrete after Seawater Freeze–Thaw Cycles

### 2.1. Theory of the CDP Model

The research on the compression characteristic of concrete materials mainly focuses on the evolution law of compressive damage to concrete. Accurately describing the compression damage of concrete materials in the process of compression could help in better analyzing the nonlinear problems of RC structures. In the process of concrete compression, the microcracks are produced and connected inside the concrete under the action of axial load, resulting in a reduction in the actual area of the concrete. Based on the above phenomena, Kachanov et al. [21] proposed taking the microcracks inside the concrete as the main factor of compression damage, and the calculation formula of a continuous variable $\Phi$ is defined as follows:

$$\Phi = \frac{A'}{A} \tag{1}$$

where $A'$ represents the actual area (the undamaged area) of concrete material. $A$ represents the nominal area (the entire area) of concrete.

The actual area of concrete decreases gradually due to the gradual development of microcracks, and the damaged area gradually increases in the process of compression. Based on Equation (1), Rabotnov et al. [22] defined the compressive damage $D_c$ of concrete during compression as the ratio of the damaged area to the nominal area, which is shown in Equation (2).

$$D_c = 1 - \Phi = 1 - \frac{A'}{A} \tag{2}$$

where $D_c$ represents the compression damage of concrete in the process of compression. ① When $D_c = 0$, concrete material has no compression damage. ② When $D_c = 1$, concrete is completely damaged under compression.

The concrete compression damage calculation model proposed by Rabotnov et al. has an intuitive physical meaning and simple mathematical formula [22] and became an essential theoretical basis for later research. Theoretically speaking, the undamaged area could be calculated by observing microcrack development inside the concrete during the process of concrete compression. However, it is difficult to observe the development of microcracks during the process of concrete compression in practical application. Hence, Lemaitre et al. [23] proposed using the degradation of reloading modulus to indirectly describe the compression damage of concrete based on the assumption of strain equivalence in the process of compression. In the uniaxial repeated compression test, when the specimen exceeds the elastic stage, the reloading modulus will gradually decrease, and the degree of reduction can be described as compression damage. The calculation formula for compression damage of concrete is as follows:

$$D_c = \frac{E_c - E_{re}}{E_c} \tag{3}$$

where $E_c$ and $E_{re}$ represent the elastic and reloading modulus (secant modulus) of concrete in the process of compression, respectively.

The study on concrete compression damage by Lemaitre et al. is based on a concrete damage plasticity (CDP) model that considers the degradation effect of the elastoplastic modulus of concrete. Isotropic elastic and compressive plasticity damage are used to replace the inelastic behavior of concrete in the CDP model, and the schematic diagram of the model is shown in Figure 3. $\varepsilon_c^{el}$ and $\varepsilon_{c0}^{el}$ represent the elastic strain with and without damage, respectively, $\varepsilon_c^{pl}$ represents the plastic strain in the process of concrete compression, and $\varepsilon_c^{in}$ represents the inelastic strain in the process of concrete compression.

The calculation method of compression damage on concrete could be resolved through uniaxial repeated compression tests on concrete prismatic specimens in the previous studies [17,24], and these tests were also adopted in this study to determine the compression damage on concrete after seawater freeze–thaw cycles. The degradation of the reloading modulus of concrete could be determined through the uniaxial repeated compression test, and the compression damage could be determined by Equation (3) based on the CDP model. Then the constitutive relationship of concrete after seawater freeze–thaw cycles could be obtained along with the concrete skeleton curve calculation method, which provides a basis for the nonlinear analysis of offshore RC structures in cold regions.

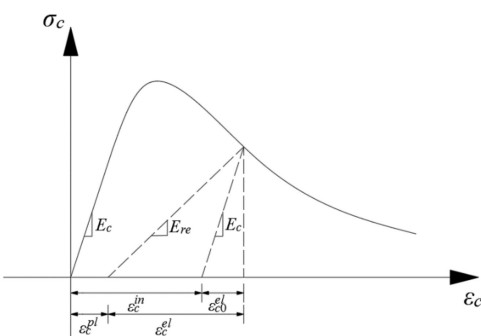

**Figure 3.** Schematic diagram of the CDP model.

### 2.2. Compression Damage of Concrete

Uniaxial repeated compression tests were carried out on concrete specimens (size 100 mm × 10 mm × 300 mm) after seawater freeze–thaw cycles, and the number of freeze–thaw cycles was 0, 25, 50, 75, 100, and 125. Freeze–thaw cycle tests and uniaxial repeated compression tests were carried out following the requirements of the code, and the mix proportions of concrete used in the test are shown in Table 1 (Section 4 presents results for the same concrete). The phenomenon of the seawater freeze–thaw cycle test is shown in Figure 4.

**Table 1.** Mix proportions of concrete (kg/m$^3$).

| Water–Cement Ratio | Cement | Fine Aggregate | Coarse Aggregate | Water | Fly Ash | Water Reducer |
|---|---|---|---|---|---|---|
| 0.48 | 373 | 873 | 838 | 180 | 66 | 8.8 |

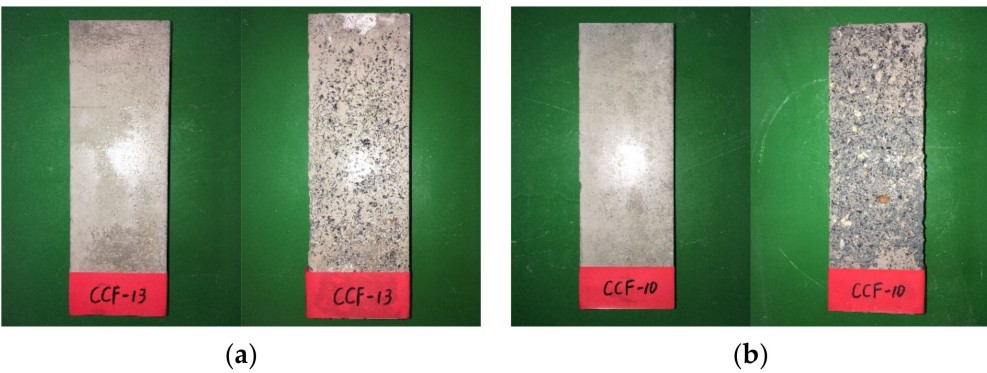

(**a**)                                                (**b**)

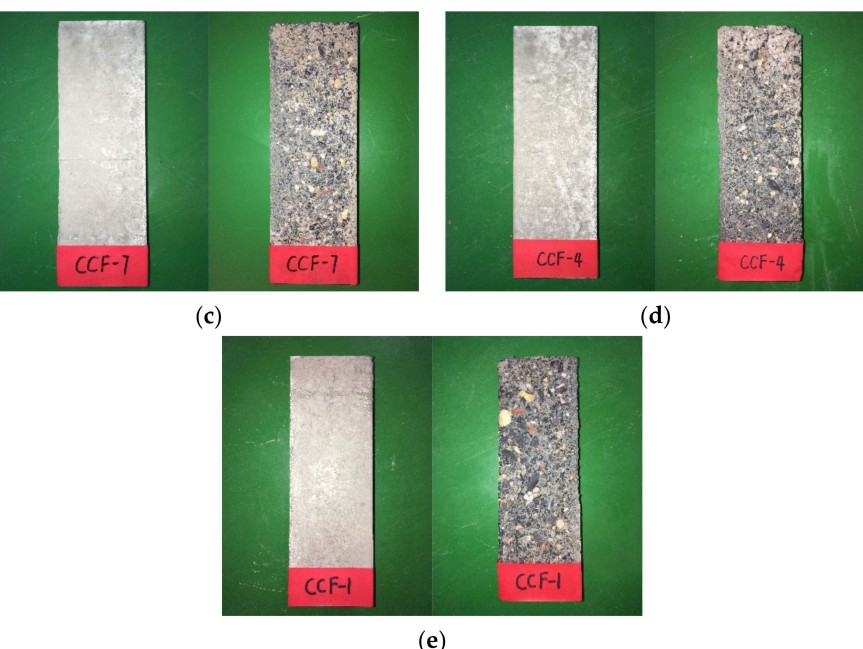

(c)　　　　　　　　　　　　　　　　　(d)

(e)

**Figure 4.** The phenomenon of seawater freeze–thaw cycle test. (**a**) Comparison before and after 25 freeze–thaw cycles. (**b**) Comparison before and after 50 freeze–thaw cycles. (**c**) Comparison before and after 75 freeze–thaw cycles. (**d**) Comparison before and after 100 freeze–thaw cycles. (**e**) Comparison before and after 125 freeze–thaw cycles.

After the rapid freeze–thaw cycle test, the concrete specimens were subjected to the uniaxial repeated compression test. A microcomputer-controlled electro-hydraulic servo universal testing machine (Mechanical Testing & Simulation, Eden Prairie, Minnesota, USA) was adopted for the uniaxial repeated compression test. The model of the testing machine was MTS/E64.206. The loading equipment is shown in Figure 5. The force sensor and displacement sensor built into the universal testing machine were used to record the load and displacement of concrete specimens in the process of repeated compression in this study. The load was applied at equal intervals at the loading rate of 0.1 mm/min. When the predetermined displacement was reached, the load was unloaded at the same rate until the load reached zero, and then loading and unloading continued at equal intervals with the new predetermined displacement as the target. The stress–strain curves of concrete specimens obtained during the uniaxial repeated compression test under different seawater freeze–thaw cycles are shown in Figure 6.

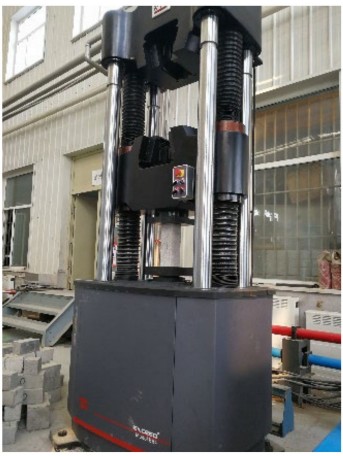

**Figure 5.** Loading equipment of electro-hydraulic servo universal testing machine.

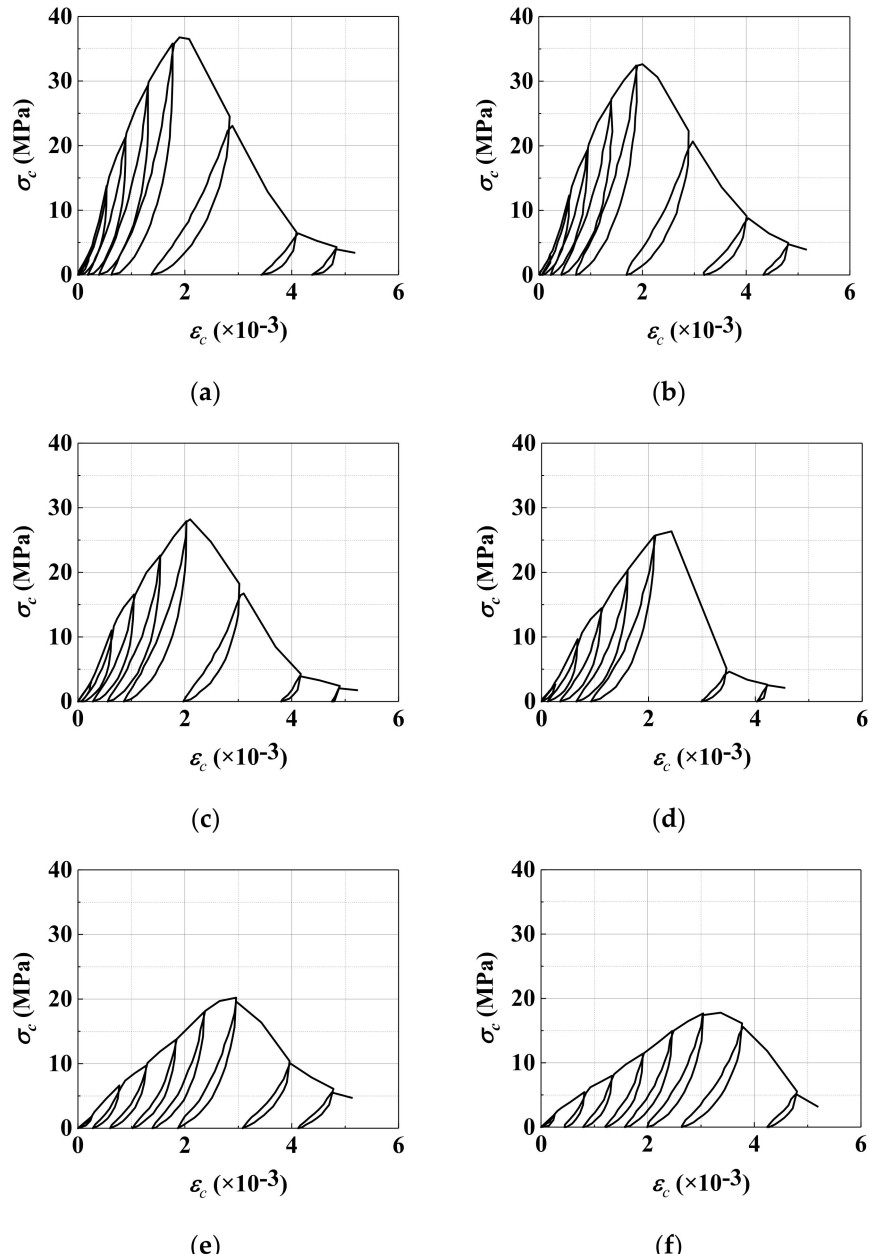

**Figure 6.** Uniaxial repeated compression test results. (**a**) $N = 0$. (**b**) $N = 25$. (**c**) $N = 50$. (**d**) $N = 75$. (**e**) $N = 100$. (**f**) $N = 125$.

The data of reloading modulus and strain in the process of the uniaxial compression test were extracted as shown in Figure 7, and different colors represent different numbers of seawater freeze–thaw cycles. It seems that the compression process could be divided into the compaction and damage stages according to the variation trend of the reloading modulus. The compression process is split into two parts by dotted lines in Figure 7:

(1) When the loading strain was less than or equal to the peak strain during the test, the concrete specimen was in the compaction stage. The internal pore structure of concrete was damaged by the seawater freeze–thaw cycles at this stage, and the loosening of the pore structure may be the leading cause of the "compression effect". In other words, the reloading modulus of concrete specimens increased with the increase in strain in the process of uniaxial repeated compression after seawater freeze–thaw cycles. The reloading modulus at the peak point of the specimen is closer to the actual elastic modulus of the concrete specimen after seawater freeze–thaw cycles. The reloading modulus at the peak

point of the specimen gradually decreases, and the peak strain increases with the increase in the number of seawater freeze–thaw cycles.

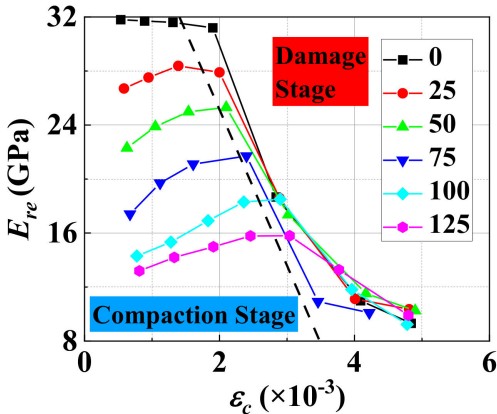

**Figure 7.** Relationship between reloading modulus and strain.

(2) When the loading strain exceeded the peak strain, the concrete specimen entered the damage stage. The reloading modulus of the concrete specimen decreased significantly after seawater freeze–thaw cycles with the increase in loading strain. The difference in the reloading modulus at the ultimate point under different freeze–thaw cycles was relatively small. The degradation trend of the reloading modulus of concrete specimens was similar under different seawater freeze–thaw cycles at the damage stage.

The curve of reloading modulus and strain after the peak point under different seawater freeze–thaw cycles in the uniaxial repeated compression test was drawn, as shown in Figure 8. Normalized processing and data fitting were carried out on the test data after the peak point, and a fitted curve was used to describe the degradation law of the reloading modulus after the peak point in the uniaxial compression test under different freeze–thaw cycles in the study of Liu [25]. The model of Liu [25] only used the test results of one strength grade of concrete for fitting, which is not universal. A degradation curve (dotted line in Figure 8) of the reloading modulus of non-freeze–thaw concrete specimens was proposed to describe the degradation trend of the reloading modulus of concrete specimens under different freeze–thaw cycles in our work. The advantages are as follows: ① The influence of the number of freeze–thaw cycles on the reloading modulus degradation law was eliminated. ② Through the degradation law of reloading modulus, the freeze–thaw damage of concrete specimens was related to the compression damage, making the conclusions of this paper more general.

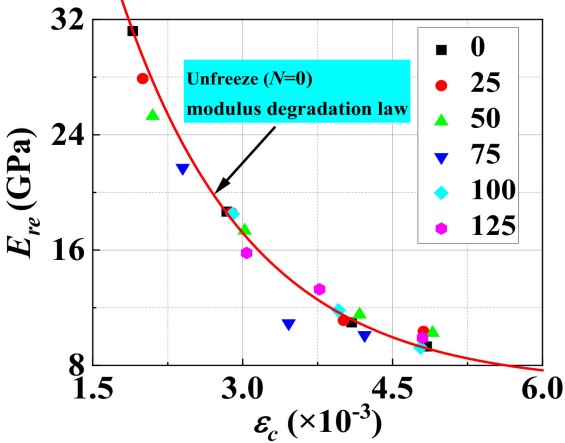

**Figure 8.** Relationship between reloading modulus and strain after peak point (seawater).



Through regression analysis of test data in Figure 8, it could be determined that the calculation formula between strain and reloading modulus of non-freeze–thaw concrete specimens in the process of uniaxial repeated compression is as follows:

$$E_{re} = 0.0256\varepsilon_c^{-1.117} \tag{4}$$

By plugging the strain value of concrete under different freeze–thaw cycles into the process of uniaxial compression into Equation (4), the corresponding reloading modulus $E_{re}$ could be determined. When the loading strain $\varepsilon_c$ is equal to the peak strain $\varepsilon_{cc}$, the loading modulus $E_{re}$ is equal to the elastic modulus $E_c$. Finally, concrete compression damage under uniaxial compression with different numbers of freeze–thaw cycles could be determined according to Equation (3).

The test data of concrete specimens in the uniaxial repeated compression test were extracted, and the relationship between the reloading modulus and strain of concrete specimens under different seawater freeze–thaw cycles in uniaxial repeated compression tests was drawn and compared with the curve described in Equation (4), as shown in Figure 8. The degradation curve of the reloading modulus of non-freeze–thaw concrete specimens could be used to describe the degradation trend of the reloading modulus of concrete specimens under different seawater freeze–thaw cycles. According to the uniaxial repeated compression test of the concrete specimen after the seawater freeze–thaw cycles, a degradation curve of the reloading modulus of non-freeze–thaw concrete specimens was used to describe the degradation law of the reloading modulus of concrete specimens under different seawater freeze–thaw cycles in this paper. In-depth analysis is needed to determine whether this law conforms to concrete with varying test environments, different strength grades, and admixtures.

The mechanical properties of concrete with different strength grades and different admixtures under uniaxial repeated compression after freshwater freeze–thaw cycles were studied by uniaxial repeated compression tests on concrete samples after freeze–thaw cycles in the experimental studies found in relevant literature [26,27]. Concrete samples with strength grades of C40 and C50 were selected as test objects in the study by Liu [26]. Recycled aggregate concrete samples with 0%, 50%, and 100% substitute recycled aggregate were selected as test objects in the study by Qi [27], and the grade of the concrete was C30. The data of concrete specimens after repeated uniaxial compression tests in [26,27] were extracted, and the relationship between the reloading modulus and strain of concrete specimens with different strength grades and admixtures after peak points was drawn, as shown in Figure 9. The data of reloading modulus are compared with those of the degradation curve of non-freeze–thaw concrete specimens in the test (dotted line in Figure 9).

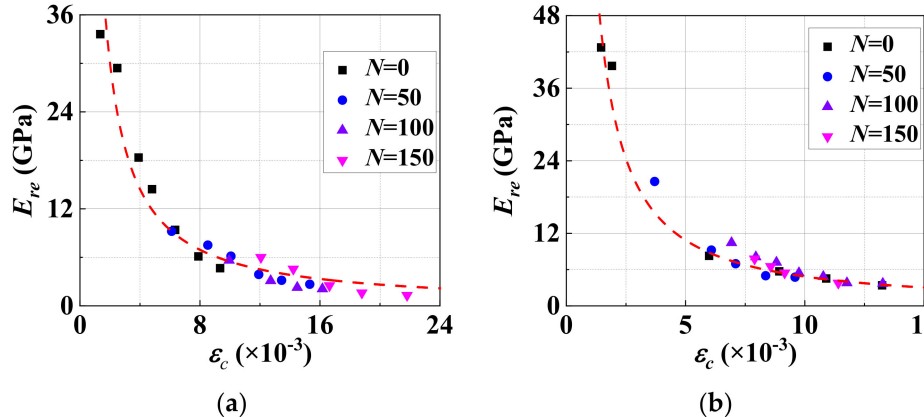

(a)  (b)

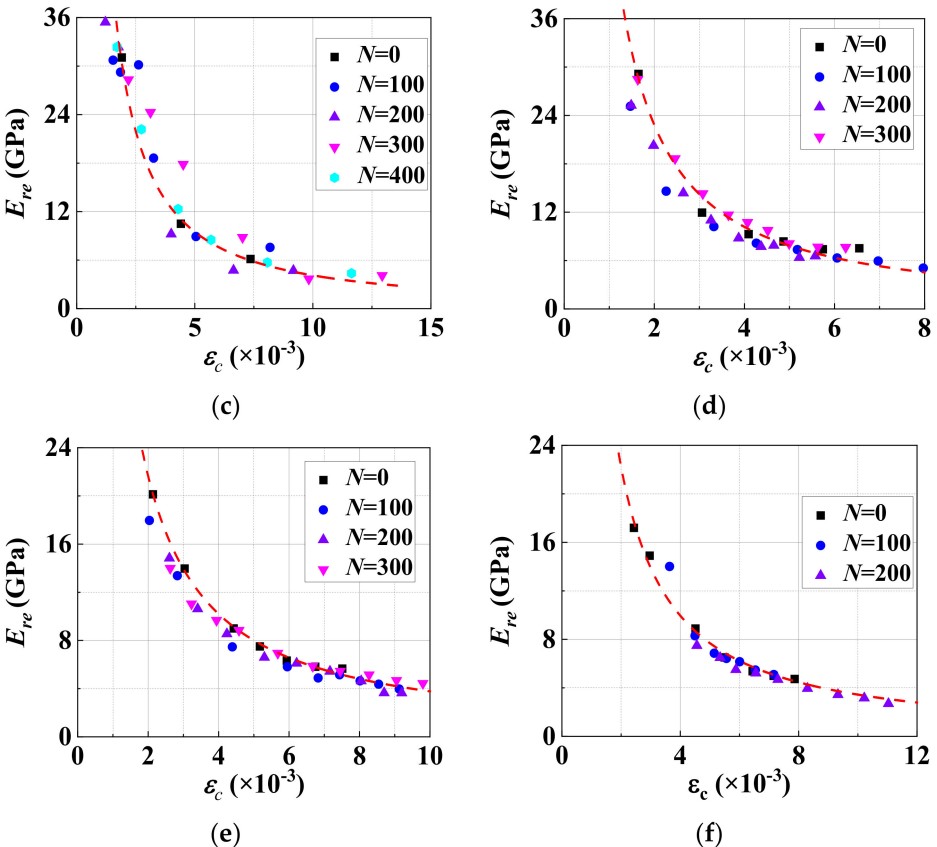

**Figure 9.** Relationship between reloading modulus and strain after peak point (freshwater). (**a**) C40 ordinary concrete [26]. (**b**) C50 ordinary concrete [26]. (**c**) C40 air-entrained concrete [26]. (**d**) C30 natural aggregate concrete [27]. (**e**) C30 recycled aggregate concrete (50% substitution) [27]. (**f**) C30 recycled aggregate concrete (100% substitution) [27].

It can be seen from Figure 9 that ① the reloading modulus of each concrete specimen decreased significantly under different freshwater freeze–thaw cycles with the increase in loading strain. ② The reloading modulus of each concrete specimen at the peak point gradually decreases, and the peak displacement increases with the increase in the number of freshwater freeze–thaw cycles. ③ The degradation law of the reloading modulus under different freeze–thaw cycles could be described by the degradation curve of non-freeze–thaw concrete, which proves the generality of the conclusion in each group of tests.

Based on the result of the freshwater and seawater freeze–thaw cycle test above, a calculation model of concrete compression damage under uniaxial compression after freeze–thaw cycles is proposed in this paper, as shown in Figure 10. The model describes the relationship between compression damage and strain of concrete under uniaxial compression after freeze–thaw cycles, as shown in Figure 10a, and the relationship between reloading modulus and strain after the peak point is shown in Figure 10b. The model is described as follows:

(1) The uniaxial compression process of concrete specimens is divided into two stages using the peak point as the reference point, as shown in Figure 10a: ① The concrete is in the compaction stage when the loading displacement does not exceed the peak displacement; with the increase in loading strain, the reloading modulus of concrete increases gradually after freeze–thaw cycles, and there is no accumulation of compression damage during this stage. ② The concrete is in the damage stage when the concrete loading exceeds the peak displacement; with the increase in loading strain, the compression damage of concrete accumulates and increases gradually.

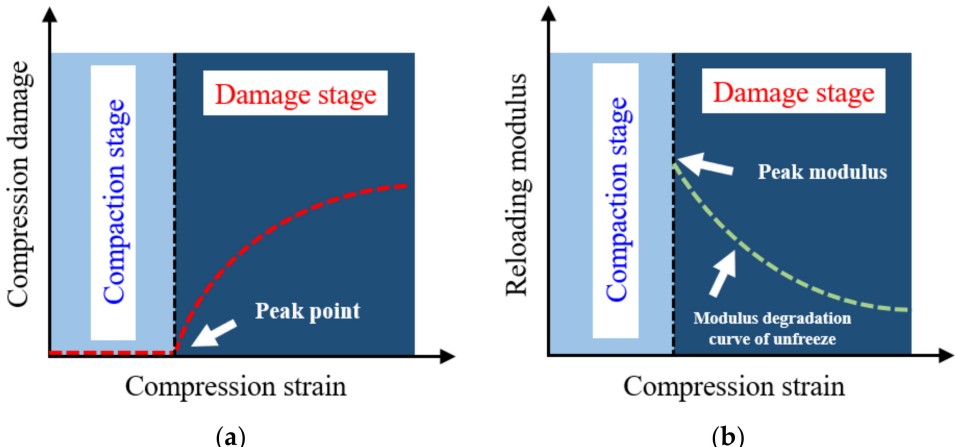

**Figure 10.** Compression damage model. (**a**) Relationship between compression damage and strain. (**b**) Relationship between reloading modulus and strain.

(2) No compression damage occurred in the concrete specimen during the compaction stage, as shown in Figure 10b. The reloading modulus at the peak point is defined as the elastic modulus of concrete after freeze–thaw cycles. When the loading displacement exceeds the peak strain, the concrete enters the damage stage. In other words, the reloading modulus starts to decrease gradually. The degradation curve of the reloading modulus on non-freeze–thaw concrete is used to describe the degradation law of the reloading modulus on concrete specimens under different freeze–thaw cycles.

*2.3. Skeleton Curve*

The expression of the skeleton curve in the Mander constitutive model [28] could well meet the geometric characteristics of the skeleton curve on concrete after freeze–thaw cycles, and the model has the characteristics of a relatively simple form and relatively few parameters at the same time. Therefore, the skeleton curve of concrete in the process of uniaxial compression after seawater freeze–thaw cycles is given based on the expression of the skeleton curve in the Mander constitutive model. Referring to previous research results [29], the function expression of the skeleton curve is as follows:

$$\frac{\sigma_c}{\sigma_{cc}} = \frac{\left(\frac{\varepsilon_c}{\varepsilon_{cc}}\right) r}{r - 1 + \left(\frac{\varepsilon_c}{\varepsilon_{cc}}\right)^r} \tag{5}$$

$$r = \frac{n E_c}{E_c - E_{re}} \tag{6}$$

where $r$ represents the shape parameter. $n$ represents the corrected parameter, and the specific form is shown in Equation (7):

$$n = 2 \times 10^{-4} N^2 + 0.01 N + 1.8 \tag{7}$$

where $N$ represents the number of seawater freeze–thaw cycles.

The uniaxial tensile constitutive relationship of concrete material after the freeze–thaw cycle can be calculated according to the uniaxial tensile constitutive relationship of ordinary concrete [18]. The uniaxial tensile constitutive relation of ordinary concrete determined in [30] is adopted in this paper, and the function expression is divided into upper and lower parts as follows:

$$\frac{\sigma_c}{\sigma_{tc}} = 1.2\left(\frac{\varepsilon_t}{\varepsilon_{tc}}\right) - 0.2\left(\frac{\varepsilon_t}{\varepsilon_{tc}}\right)^6 \qquad 0 \le \frac{\varepsilon_t}{\varepsilon_{tc}} \le 1 \tag{8}$$

$$\frac{\sigma_c}{\sigma_{tc}} = \frac{\left(\frac{\varepsilon_t}{\varepsilon_{tc}}\right)}{0.312\sigma_{tc}^2 \left(\frac{\varepsilon_t}{\varepsilon_{tc}} - 1\right)^2 + \frac{\varepsilon_t}{\varepsilon_{tc}}} \qquad \frac{\varepsilon_t}{\varepsilon_{tc}} \geq 1 \tag{9}$$

where $\sigma_t$ and $\varepsilon_t$ represent the stress and strain in concrete during uniaxial tension, respectively. $\sigma_{tc}$ and $\varepsilon_{tc}$ represent the peak stress and peak strain during uniaxial tension of the concrete, respectively.

According to [18], the calculation formula of concrete uniaxial tensile peak stress is determined as follows:

$$\sigma_{tc} = 0.625\sqrt{\sigma_{cc}} \tag{10}$$

The damage to concrete during uniaxial tension can be calculated by Equation (11) [18]:

$$D_t = 1 - \frac{\sigma_t E_t^{-1}}{\varepsilon_t^{in}\left(\frac{1}{b_t} - 1\right) + \sigma_t E_t^{-1}} \tag{11}$$

where $\varepsilon_t^{in}$ represents the tensile inelastic strain of concrete, and $\varepsilon_t^{in} = \varepsilon_t - \sigma_t/E_t$. $E_t$ represents the tensile elastic modulus of concrete and $E_t = E_c$. $b_t$ represents the ratio of tensile plastic strain to the inelastic strain of concrete, according to [31], and the value is 0.7.

## 3. Equivalent Meso-Element Model

### 3.1. Theory of Meso-Element Equivalence

Concrete is a porous material with a random distribution of coarse aggregate inside, and the random distribution is also known as the heterogeneity of concrete materials [32]. The heterogeneity of concrete material makes its macroscopic mechanical properties show strong discreteness, and the dispersion of concrete material is more evident after seawater freeze–thaw cycles [33]. It is necessary to accurately describe the heterogeneity of the concrete to truly reflect its nonlinear mechanical properties.

Building a random aggregate model of concrete at a mesoscale could truly reflect the relationship between mesoscopic damage and macroscopic mechanical properties of concrete materials in the numerical simulation process [34]. However, the numerical simulation study of nonlinear problems on three-dimensional concrete structures with the random aggregate model would seriously reduce the computational efficiency of the numerical simulation process. The equivalent meso-element model was proposed in [32] to describe the heterogeneity of concrete based on a random aggregate model of concrete to improve the computational efficiency of the numerical simulation process. Based on the random aggregate model, establishing an equivalent meso-element model by an equivalent method of composite material was called the meso-element equivalence method. The advantage of using an equivalent meso-element model in numerical simulation is that it reduces the influence of concrete structure due to the size effect while considering the heterogeneity of concrete material [32].

On the mesoscopic scale, concrete can be understood as a composite material, and its interior is composed of coarse aggregate and cement mortar. By setting the shape, size, and distribution form of coarse aggregate, a concrete random aggregate model similar to natural concrete material can be established, as shown in Figure 11a. The same size of the element is used to divide the concrete random aggregate model into equal intervals, and the volume fraction of coarse aggregate $C_{ag}$ and cement mortar $C_{mo}$ in each element was different. Given the elastic modulus $E_{ag}$ of coarse aggregate and $E_{mo}$ of cement mortar, the calculation formula of elastic modulus $E_c$ of each concrete element is as follows [32]:

$$E_c = E_{ag}C_{ag} + E_{mo}C_{mo} \tag{12}$$

where $E_c$ represents the elastic modulus of concrete, $E_{ag}$ and $E_{mo}$ represent the elastic modulus of coarse aggregate and mortar, and $C_{ag}$ and $C_{mo}$ represent the volume fraction of coarse aggregate and mortar.

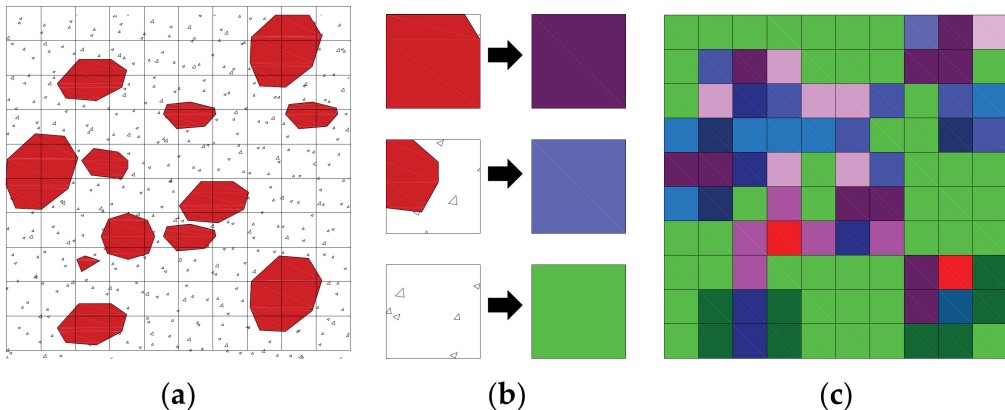

**Figure 11.** Method of meso-element equivalence. (**a**) Random aggregate model. (**b**) Equivalent process. (**c**) Equivalent meso-element model.

The mechanical properties of each element can be treated equivalently by using the method of meso-element equivalence so that each element is equivalent to an isotropic homogeneous medium, and the equivalent process is shown in Figure 11b. The mechanical properties of each concrete element were the same in the equivalent meso-element model, but the mechanical properties between concrete elements are different. Through the method of meso-element equivalence, the equivalence meso-element model of concrete material is established [32], as shown in Figure 11c. The mechanical properties (with different colors), such as elastic modulus and strength, of different elements are different. In a comparison of the meso-element equivalent model and random aggregate model of concrete, the two models show the same heterogeneity, so the macroscopic mechanical properties of concrete materials described are the same.

In the equivalent meso-element model, the mechanical properties of concrete elements are highly discrete. The elastic modulus $E_c$ and compressive strength $\sigma_{cc}$ conform to the two-parameter Weibull distribution [32], and the expression is shown in Equation (13):

$$F(x) = 1 - \exp\left[1 - \left(\frac{x}{\beta}\right)^m\right] \tag{13}$$

The probability density function is as follows:

$$f(x) = \frac{dF(x)}{dx}\frac{m}{\beta}\left(\frac{m}{\beta}\right)^{m-1}\exp\left[1 - \left(\frac{x}{\beta}\right)^m\right] \tag{14}$$

where $\beta$ and $m$ represent the scale parameters and shape parameters of the function, respectively.

### 3.2. Heterogeneity of Concrete after Seawater Freeze–Thaw Cycles

The first step of establishing an equivalence meso-element model of concrete after seawater freeze–thaw cycles is to establish a random aggregate model of concrete. The mechanical properties of concrete specimens are related to the size of the specimens, which determines the number of internal defects of concrete, thus affecting the mechanical properties of the concrete. The size of the element in the model also affects the macroscopic mechanical properties of the concrete. According to the research results of [32], the random aggregate model of two-phase (coarse aggregate and cement mortar) concrete was reasonably selected in this study. The distribution of coarse aggregate inside concrete material satisfies the Fuller grading curve, which could effectively describe the compactness and

macroscopic mechanical properties of concrete material [34]. The expression of the Fuller grading curve is shown in Equation (15):

$$p(d) = 100 \left( \frac{d}{d_{\max}} \right)^{\kappa} \tag{15}$$

where $p(d)$ represents the volume fraction of coarse aggregate in which the particle size is smaller than $d$. $d_{\max}$ represents the maximum particle size of coarse aggregate. $\kappa$ represents the shape parameter of the function, and the value is 0.5 according to [32].

The volume fraction of coarse aggregate particles in the random aggregate model is determined to be 40%, and that of cement mortar is 60% according to the mix ratio of the concrete used in this study. Based on the cross-section of the standard concrete elastic modulus test specimen (size 150 mm × 150 mm), the random aggregate model of the concrete specimen was established in MATLAB, as shown in Figure 12. The circular areas represent coarse aggregate particles (the color blue), and the remaining areas represent the cement mortar matrix in the random aggregate model (the color yellow). Fifteen millimeters was reasonably selected as the size of the feature element [32], and the random aggregate model was meshed as shown in Figure 12.

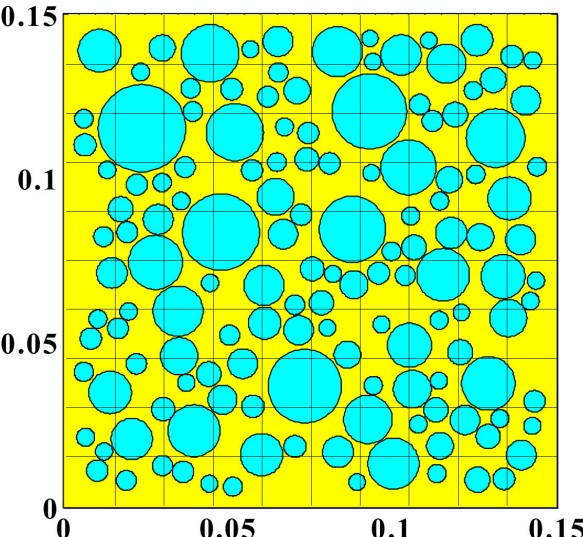

**Figure 12.** Random aggregate model of concrete (unit: m).

The elastic modulus $E_m$ of concrete in each element is determined according to Equation (5) in this paper. It is necessary to determine the elastic modulus $E_{ag}$ of coarse aggregate and the elastic modulus $E_{mo}$ of cement mortar after seawater freeze–thaw cycles firstly. There is no separation and relative sliding between coarse aggregate and cement mortar, and the ideal interface state is always maintained when concrete deforms under freeze–thaw cycles and loads. Therefore, the elastic modulus of coarse aggregate $E_{ag}$ remains unchanged during freeze–thaw cycles [35]. According to [32], the elastic modulus of the coarse aggregate used in this study is 50 GPa, and its value stays constant invariably. The elastic modulus $E_{mo}$ of cement mortar needs to be calculated. Finite element analysis software ABAQUS was used to calculate the elastic modulus $E_{mo}$ of cement mortar inside concrete after seawater freeze–thaw cycles in this study. The location coordinates and radius size of the coarse aggregate were extracted based on the random aggregate model of concrete specimens established in MATLAB, and Python commands were used to read the coordinates and radius size of the aggregate. Then a random aggregate model was established in ABAQUS as shown in Figure 13.

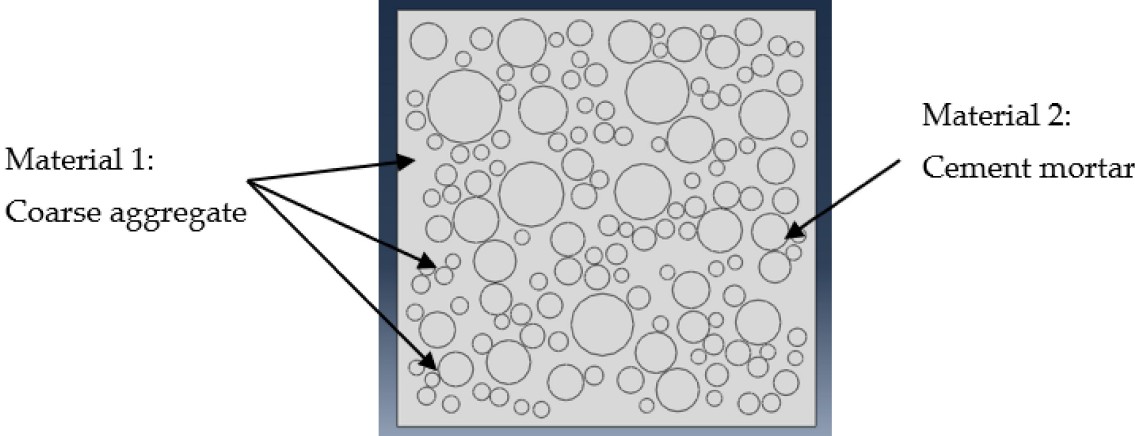

**Figure 13.** Random aggregate model of concrete in ABAQUS.

The elastic modulus $E_{mo}$ of cement mortar in concrete after the seawater freeze–thaw cycle was calculated by the inversion method in this study, and the basic idea is shown in Figure 14 and described as follows: ① Through uniaxial compression tests on concrete specimens after seawater freeze–thaw cycles, the elastic modulus $E_m$ of concrete specimens under different seawater freeze–thaw cycles was determined, as shown in Table 2. ② The elastic modulus of coarse aggregate $E_{ag}$ = 50 GPa and the (estimated) initial value of elastic modulus of cement mortar $E'_{mo}$ = 30 GPa were input into the software. ③ The numerical model was loaded by uniaxial compression load, and the elastic modulus $E'_m$ of the concrete numerical model was determined in the software ABAQUS. ④ The elastic modulus of cement mortar $E_{mo}$ in concrete was calculated by the inversion method, for which the iteration method was the Newton iteration method, and the convergence condition was $(E_m - E'_m)/E_m \leq 5\%$. ⑤ The elastic modulus $E_{mo}$ of cement mortar in concrete after the freeze–thaw cycles was determined. The elastic modulus values of cement mortar $E_{mo}$ under different seawater freeze–thaw cycles were determined in this study, as shown in Table 2.

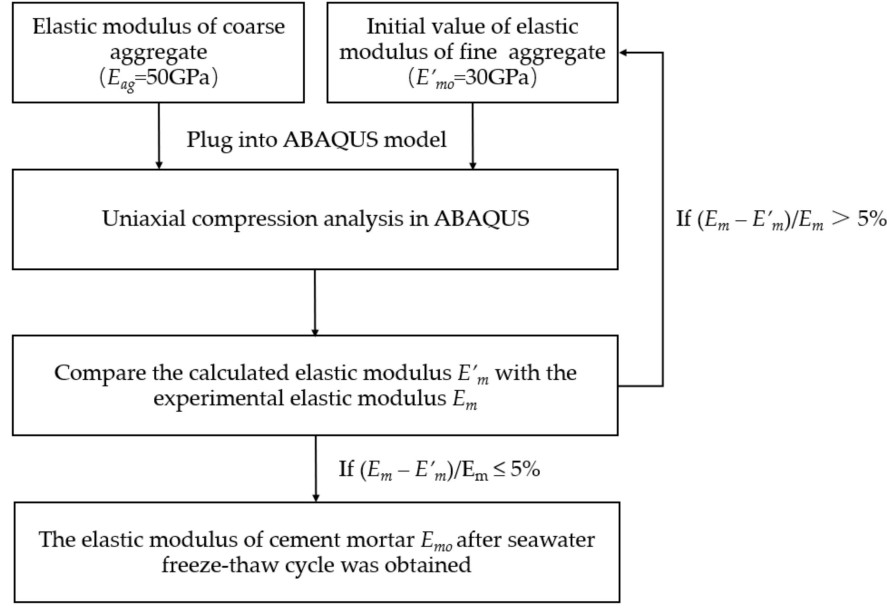

**Figure 14.** Analysis procedure for the elastic modulus values of cement mortar Emo.

**Table 2.** The elastic modulus of concrete and mortar.

| N | $E_m$(GPa) | $E_{mo}$ (GPa) |
|---|---|---|
| 0 | 31.2 | 22.8 |
| 25 | 27.9 | 18.6 |
| 50 | 25.3 | 16.5 |
| 75 | 21.7 | 13.3 |
| 100 | 18.5 | 11.2 |
| 125 | 15.8 | 8.5 |

The program embedded in the software AutoCAD was used to calculate the volume fractions $C_{ag}$ and $C_{mo}$ of coarse aggregate and cement mortar in each element of the random aggregate model in this study. The elastic modulus $E_m$ of each concrete element in the equivalent meso-element model was calculated according to Equation (5). The elastic modulus of 100 concrete elements was analyzed mathematically, and the frequency histogram of elastic modulus was drawn with the appropriate range of intervals, as shown in Figure 15.

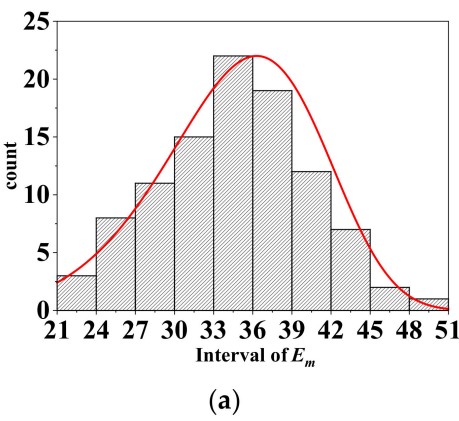

(**a**)

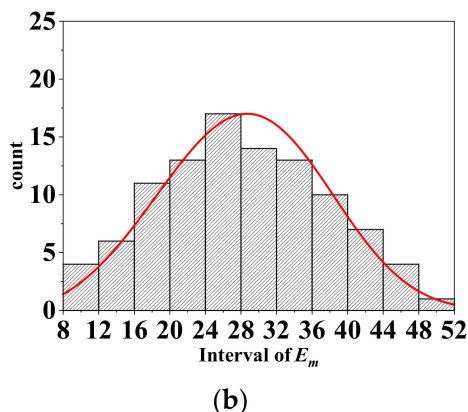

(**b**)

**Figure 15.** Histogram of frequency. (**a**) *N* = 0. (**b**) *N* = 125.

The two-parameter Weibull distribution was used to fit the probability distribution of the elastic modulus frequency histogram in each element under different seawater freeze–thaw cycles in this study. The values of parameters $\beta$ and $m$ in the two-parameter Weibull distribution under different seawater freeze–thaw cycles were determined. The results are shown in Table 3.

**Table 3.** Double parameter values of Weibull distribution.

| N | 0 | 25 | 50 | 75 | 100 | 125 |
|---|---|---|---|---|---|---|
| $\beta$ | 37.2 | 34.0 | 32.7 | 31.4 | 30.2 | 28.3 |
| $m$ | 6.5 | 5.6 | 4.6 | 3.8 | 3.3 | 3.1 |

Regression analysis was conducted on the test data to obtain the calculation formula between the number of seawater freeze–thaw cycles *N*, scale parameter $\beta$, and shape parameter m, as shown in Equations (16) and (17):

$$\beta = 36.42 - 0.07N \tag{16}$$

$$m = 6.61 - 0.05N + 1.64N^2 \tag{17}$$

## 4. Experiments and Results

### 4.1. Experiments

The design detail of the specimens is shown in Figure 16. The rapid freeze–thaw cycle test device for concrete was modified to perform a rapid freeze–thaw cycle test on the fabricated segments, as shown in Figure 17. According to the size of the abdominal cavity of the test device, in this study, five stainless steel containers were designed and manufactured, as shown in Figure 17a; the size of each container was 280 mm × 240 mm × 510 mm, and the containers were used for the freeze–thaw section in the rapid freeze–thaw cycle test. All four sides of the freeze–thaw container section exceeded the dimensions of the freeze–thaw section by 10 mm, allowing the freeze–thaw segments to be immersed evenly in seawater. At the same time, four stainless steel containers with relatively small cross sections were designed to hold the temperature measurement specimens shown in Figure 17b; the size of the temperature measuring vessel was 120 mm × 120 mm × 510 mm.

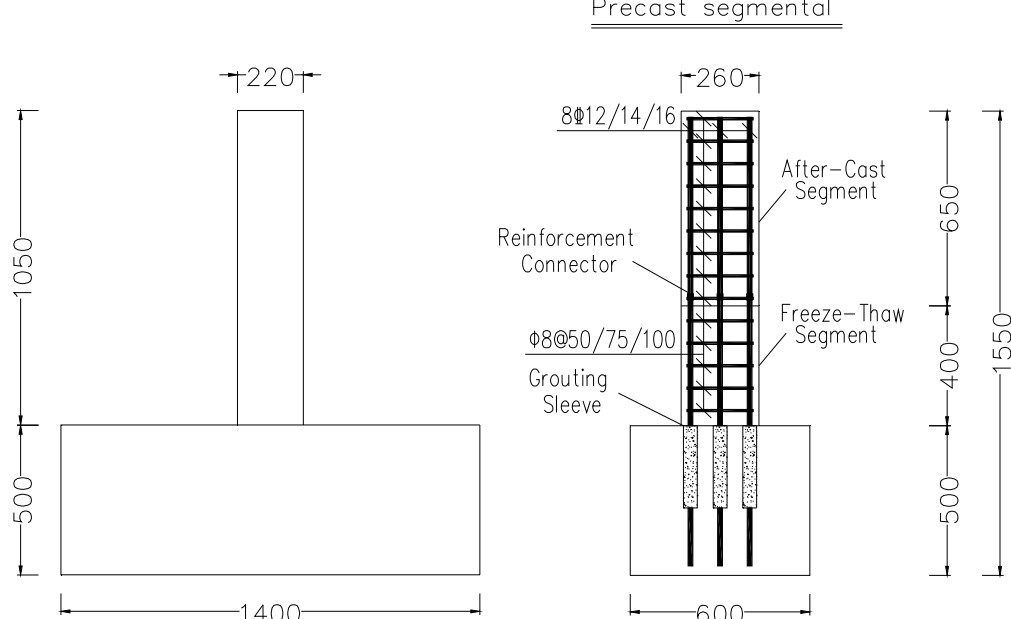

**Figure 16.** Design detail of specimens.

The concrete mix was the same as that described in Section 2, as shown in Table 1. The basic parameters of the rebar are shown in Table 4.

**Table 4.** Parameters of the longitudinal reinforcement and stirrup.

| Parameter | Longitudinal Reinforcement | Stirrup |
| --- | --- | --- |
| Reinforcement type | HRB400 | HPB300 |
| Physical quality | Ribbed | Plain |
| Diameter (mm) | 12, 14, or 16 | 8 |
| Yield strength (MPa) | 422 | 450 |
| Ultimate strength (MPa) | 605 | 550 |
| Elongation (%) | 7.2 | 10.4 |

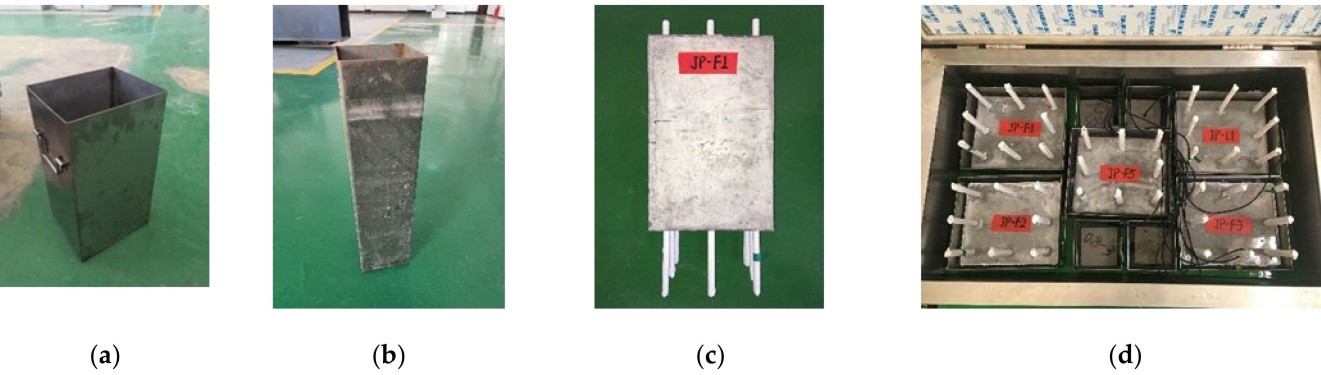

(**a**)          (**b**)          (**c**)          (**d**)

**Figure 17.** Rapid freeze–thaw cycle test for the F-T segments. (**a**) Freeze–thaw cycle container. (**b**) Constant temperature container. (**c**) Rust prevention. (**d**) Inside the testing machine.

The thawing and freezing temperatures of the concrete rapid freeze–thaw cycle testing machine were set to 5 °C and −20 °C, and the test was started. The time of a single freeze–thaw cycle was not more than 4 h. The thawing time of the specimen accounted for less than one-fourth of the time of a single freeze–thaw cycle. Taking 25 freeze–thaw cycles as the test cycle, the segments were taken out after the freeze–thaw cycles. The results of the seawater freeze–thaw cycle test on each segment are shown in Figure 18.

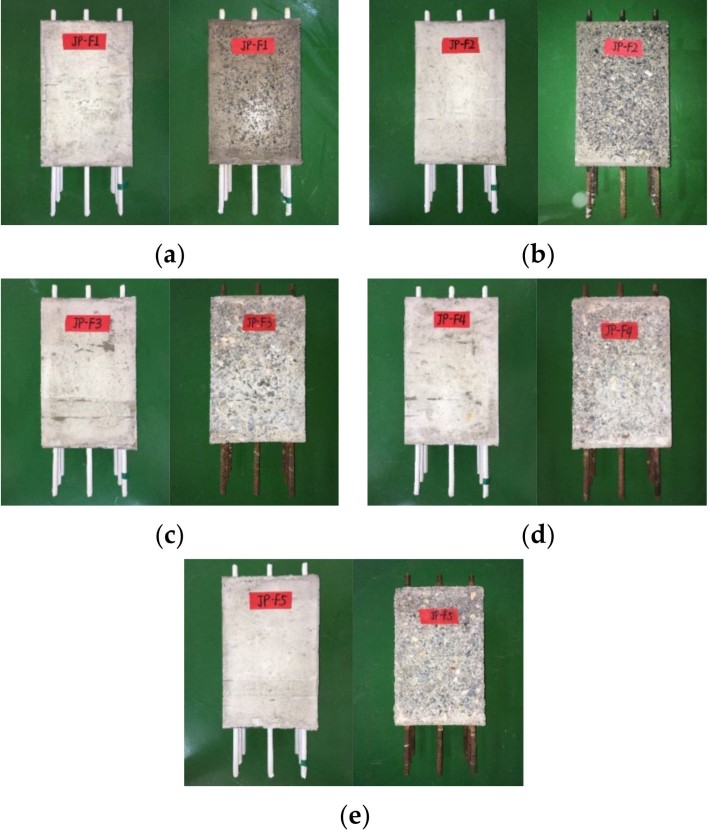

**Figure 18.** Experimental results of the rapid freeze–thaw cycles. (**a**) Comparison before and after 25 freeze–thaw cycles. (**b**) Comparison before and after 50 freeze–thaw cycles. (**c**) Comparison before and after 75 freeze–thaw cycles. (**d**) Comparison before and after 100 freeze–thaw cycles. (**e**) Comparison before and after 125 freeze–thaw cycles.

After the seawater freeze–thaw cycle test, the assembly and the secondary pouring work were carried out on the specimens, as shown in Figures 19 and 20.

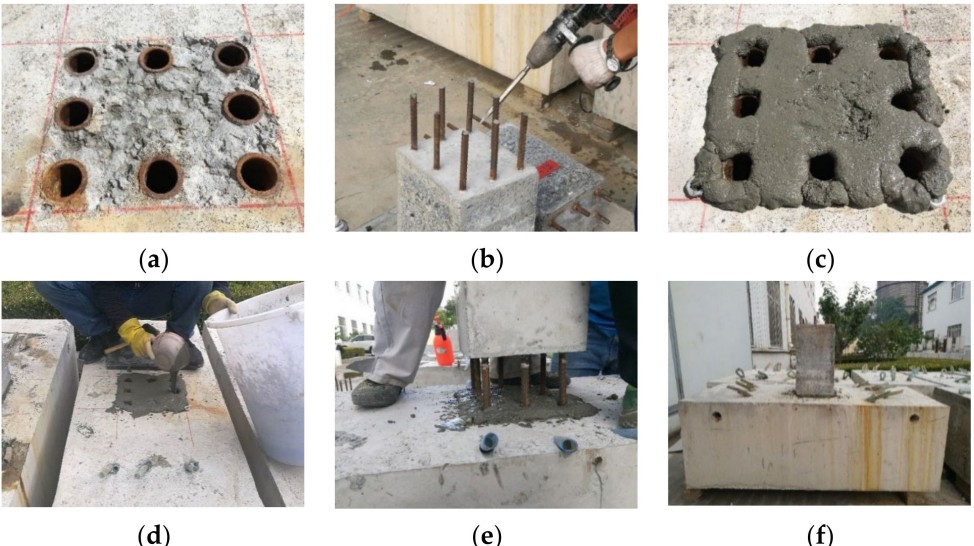

**Figure 19.** The main process of assembly. (**a**) Base cut hair. (**b**) Roughening of the F-T segment. (**c**) Laying the sealing material. (**d**) Artificial grouting. (**e**) Inserting freeze–thaw segments. (**f**) End of the assembly.

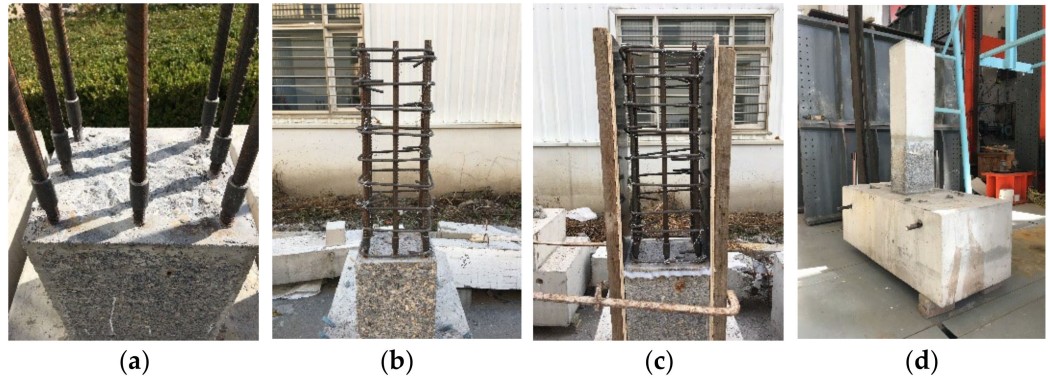

**Figure 20.** The main process of the second pouring. (**a**) Lengthening of the longitudinal bar. (**b**) Assembling reinforcement. (**c**) Formwork. (**d**) Finished pouring.

After the assembly and secondary pouring work, the low cyclic loading test was carried out in the structural laboratory of the bridge and tunnel R&D base of the Dalian University of Technology, and the test loading device is shown in Figure 21. The base of the specimen was fixed on the ground of the laboratory by two beams and four anchor bolts in the vertical direction, and the base of the specimen was fixed on the shear wall by another two beams and two long bolts in the horizontal direction. The vertical axial force was provided by a hydraulic jack with an ultimate load of 3000 kN through the spherical hinge and rolling guide rail transmission so that the specimen maintained a constant vertical axial force. The vertical hydraulic jack was controlled by the electro-hydraulic servo system with an ultimate displacement of $\pm 300$ mm and a range of $\pm 1000$ kN, ensuring that the error between the applied vertical axial force and the input value was not more than $\pm 5\%$. The axial force ($F$) can be calculated by Equation (18) according to standard JTG 3362-2018 [36].

$$F = u f_{cd} A_c \tag{18}$$

where $u$ is the designed axial compression ratio of the specimen; $f_{cd}$ is the concrete design strength; $A_c$ is the cross-sectional area of the concrete specimen.

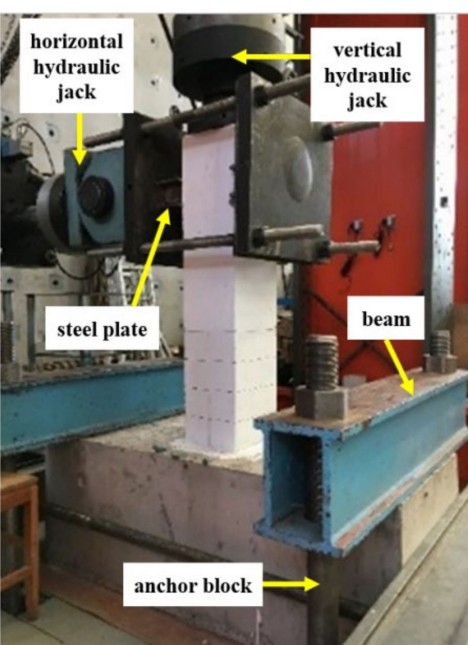

**Figure 21.** The specimen and the testing apparatus.

The specimen was subjected to displacement-controlled lateral loading, and the loading procedure is presented in Figure 22. The loading rate for the lateral load was 0.1 mm/min according to the requirements of the standard GB/T 50152-2012 [37]. When the lateral displacement amplitude was lower than the predicted yield displacement $\Delta y$ (approximately equal to ±6 mm), the lateral displacements were ±2 mm and ±4 mm. When the lateral displacement amplitude was above the predicted displacement, the loading step was multiplied based on the yielding displacement (i.e., $0.5\Delta y$, $\Delta y$, $2\Delta y$, $3\Delta y$, etc.). Three fully reverse cyclic loadings were performed at each displacement step according to the specification given in the standard JGJ/T 101-2015 [38]. The test was stopped when the lateral load dropped to 85% of the maximum lateral load.

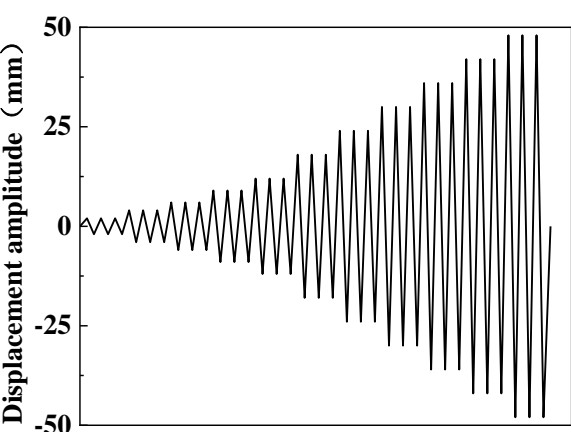

**Figure 22.** Loading procedure for the low cyclic loading test.

### 4.2. Results of the Experiments

The final failure mode diagram and displacement–load curve of the specimens (JP-F0, JP-F1, JP-F2, JP-F3, JP-F4, and JP-F5) are shown in Figures 23 and 24. JP is the Chinese abbreviation for segmentary assembly, and F0–F5 represent 0, 25, 50, 75, 100, and 125 seawater freeze–thaw cycles, respectively.

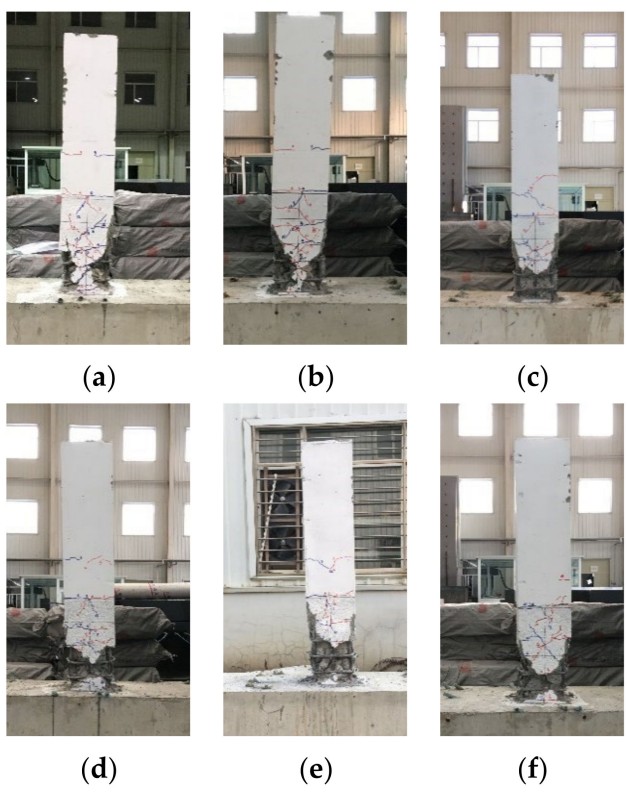

**Figure 23.** The failure mode of the specimens. (**a**) JP-F0. (**b**) JP-F1. (**c**) JP-F2. (**d**) JP-F3. (**e**) JP-F4. (**f**) JP-F5.

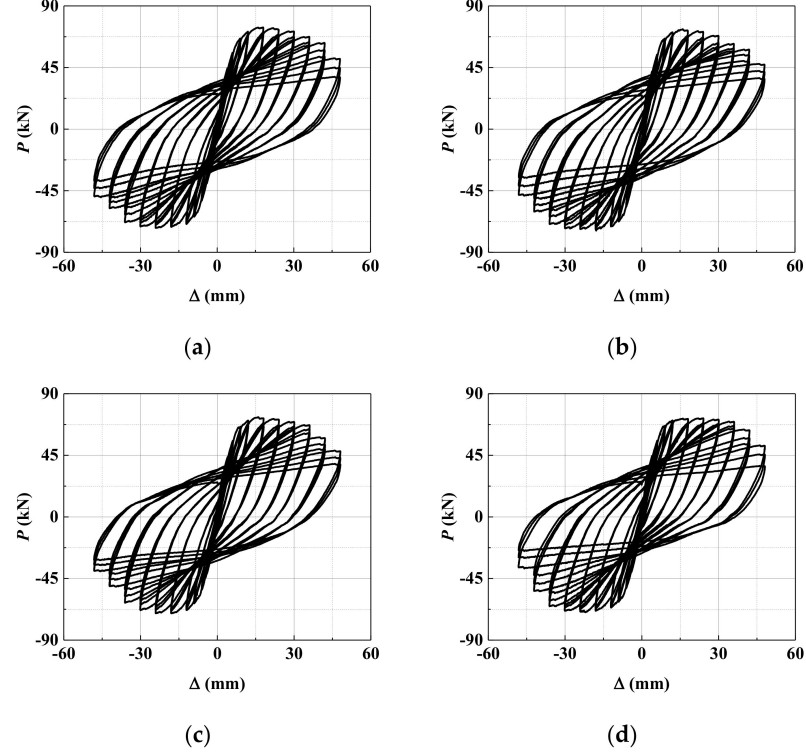

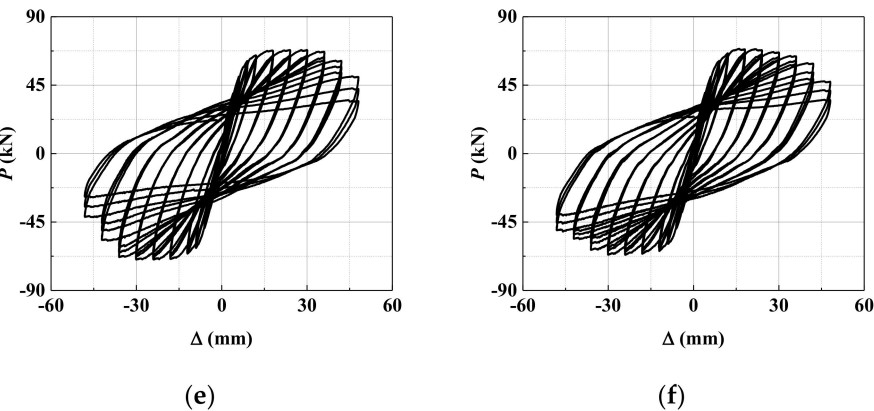

(e)

(f)

**Figure 24.** Hysteresis characteristics of the specimens. (**a**) JP-F0. (**b**) JP-F1. (**c**) JP-F2. (**d**) JP-F3. (**e**) JP-F4. (**f**) JP-F5.

## 5. Numerical Analysis Model and Results

### 5.1. Verification of Numerical Simulation

Finite element analysis software ABAQUS was used to conduct a numerical simulation of the low cyclic loading test of an RC pier after seawater freeze–thaw cycles. The effectiveness of the numerical simulation method was verified by comparing the numerical and experimental results of RC piers with different design parameters. The modeling process is shown in Figure 25: ① The C3D8R element was chosen as the concrete element. Core concrete was input according to the non-freeze–thaw concrete constitutive model, and the input of freeze–thaw-damaged concrete was as described in the previous sections; ② The beam element was chosen as the longitudinal bar and the stirrup element; the section was the actual section area of the steel bar. The mechanical properties of the rebar were input according to Table 4. ③ Conditions for binding between steel bars and concrete were embedded. ④ A reference point was established at the vertex of the model for loading the specimen, and the loading mode was expanded as shown in Figure 14. ⑤ The vertical concentration force was also exerted on the reference point with the direction pointing to the ground, and its magnitude was calculated according to Equation (18). ⑥ The boundary condition of the model was a ground-fixed constraint, which was used to simulate ground anchors.

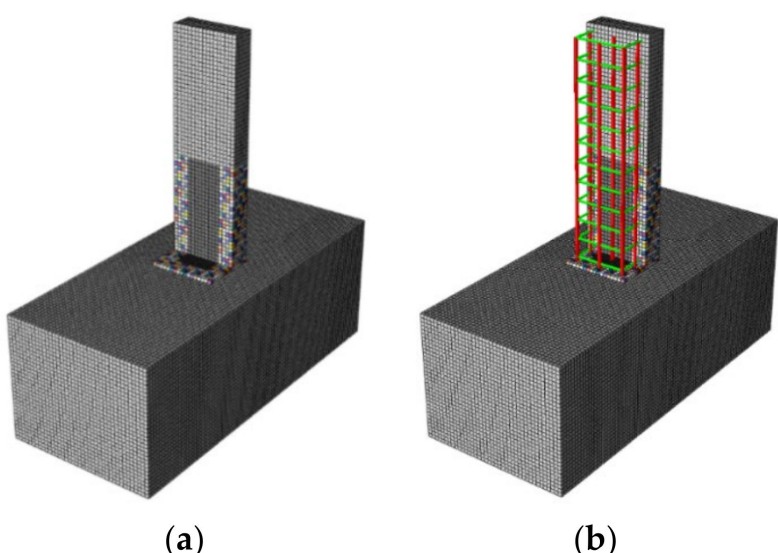

(a)

(b)

**Figure 25.** Finite element model of RC pier. (**a**) Analysis model of the pier without bars. (**b**) Analysis model of the pier with bars.

According to previous research results [39], there was no freeze–thaw damage in the whole section of the RC piers, and the freeze–thaw depth was related to the number of seawater freeze–thaw cycles suffered by the RC piers. To make the numerical model of the establishment closer to the real situation, the RC piers after seawater freeze–thaw cycles were modeled, and the layered method is shown in Figure 26. The freeze–thaw depth hu of RC piers after seawater freeze–thaw cycles was assigned according to the results calculated in previous research [39], as shown in Figure 27.

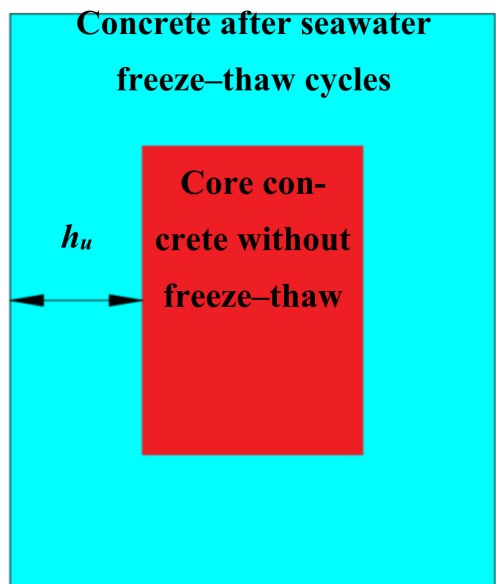

**Figure 26.** Hierarchical method.

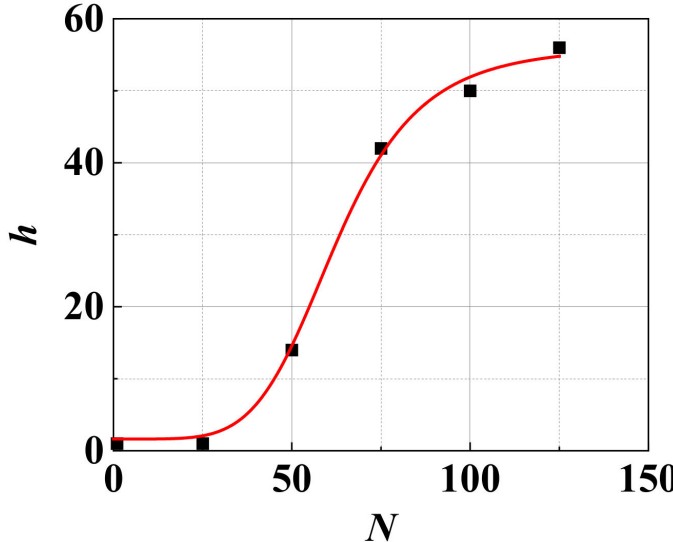

**Figure 27.** Relationship between freeze–thaw times and depth.

The low cyclic loading test of RC piers after seawater freeze–thaw cycles was numerically simulated using finite element analysis software ABAQUS; the flowchart of the whole numerical analysis process is shown in Figure 28.

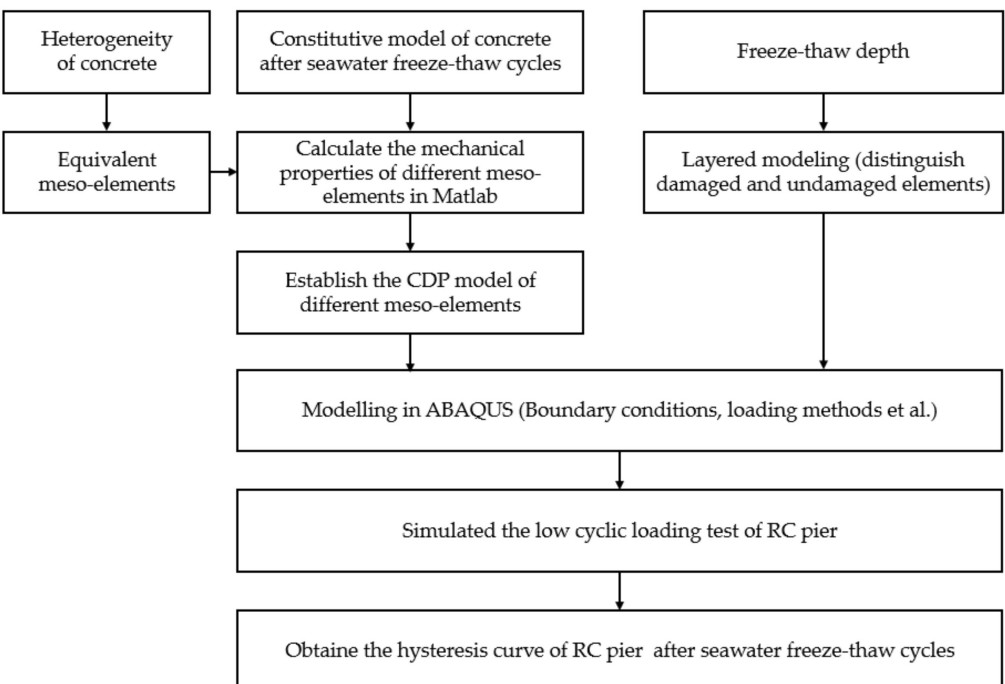

**Figure 28.** Flowchart of the whole numerical analysis process.

The comparison between simulation results and experimental results is shown in Figure 29. The hysteretic curves obtained by numerical simulation were full, and the pinching effect was good, which could reflect the various stages of the low cyclic loading test on RC piers after seawater freeze–thaw cycles:

(1) The RC pier was in an elastic stage before the longitudinal reinforcement yielded, and the deformation of the pier was relatively small at this time. The loading and unloading paths of the hysteretic curve were approximately linear, and there was no significant accumulated energy dissipation. The intersections of the unloading paths and *X*-axis are called residual deformation, which was relatively small and approximately negligible at the elastic stage.

(2) The RC pier gradually changed from the elastic stage to the plastic stage with the increase in loading. The tensile strain of reinforcement and the compressive strain of concrete gradually accumulated after the yield point, and the area enclosed by the hysteresis loop increased accordingly. The residual deformation of the pier decreased gradually with the increase in the number of seawater freeze–thaw cycles. The slope of the loading and unloading paths decreased, and the rate of decrease gradually increased with the increase in loading displacement amplitude. Under the same displacement amplitude, the slope of the loading and unloading path of the hysteretic loop gradually decreased with the loading times. The stiffness of the RC pier was gradually degraded under the action of a low cyclic loading test.

(3) The peak load $P_m$ of the RC pier in the low cyclic loading test was negatively correlated with the number of seawater freeze–thaw cycles, while the peak displacement $\Delta_m$ of the RC pier was the opposite. The cover concrete near the bottom of the pier reached the ultimate strain, which exited the operation, resulting in the longitudinal bars beginning to buckle, and the load on the top of the RC pier began to decrease.

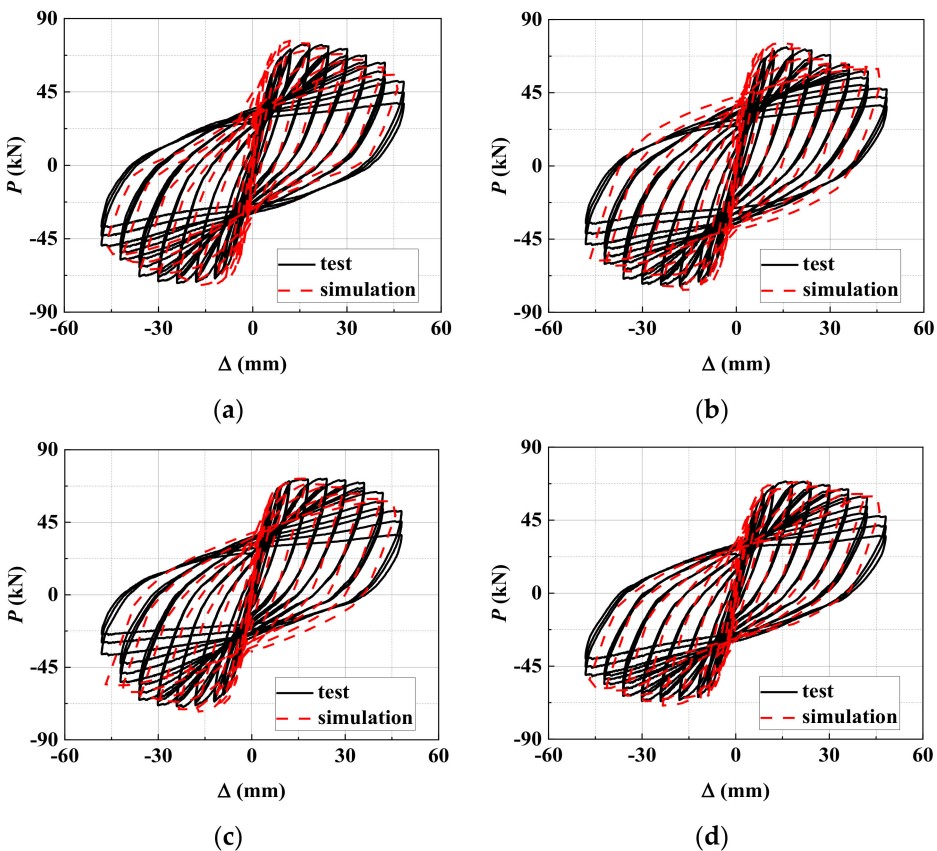

**Figure 29.** Comparison between numerical simulation and test results. (**a**) JP-F0. (**b**) JP-F1. (**c**) JP-F3. (**d**) JP-F5.

The peak load $P_m$ and peak displacement $\Delta_m$ of the numerical simulation hysteresis curve were extracted and compared with those obtained in the test, as shown in Table 5. The deviation value of peak load was not more than 6%, and the deviation value of peak displacement was not more than 10%, which is within the ideal range. It can be seen that the numerical analysis method proposed in this paper was effective.

**Table 5.** Peak load and peak displacement.

| No. | $P_m$ | | | $\Delta_m$ | | |
|-----|-----------|-------------------|----------------|-----------|-------------------|----------------|
| | Test (kN) | Simulation (kN) | Deviation (%) | Test (mm) | Simulation (kN) | Deviation (%) |
| JP-F0 | 74.2 | 73.4 | 1.1 | 18 | 17.0 | 5.6 |
| JP-F1 | 73.0 | 74.1 | −1.5 | 18 | 17.8 | 1.1 |
| JP-F3 | 69.7 | 70.2 | −0.7 | 24 | 23.3 | 2.9 |
| JP-F5 | 66.4 | 65.8 | 0.9 | 30 | 26.8 | 10.0 |

*5.2. Parameter Analysis*

The effective numerical simulation method proposed in this paper was used to analyze the parameters of the low cyclic loading test, and the parameters of analysis include axial compression ratio (0.225 and 0.3), longitudinal reinforcement diameter (12 and 16 mm), and stirrup spacing (50 and 100 mm). The hysteresis curves of specimens with different design parameters subjected to 125 seawater freeze–thaw cycles are shown in Figure 30. The letters A, L, and J are the abbreviations for the ratio of axial compression, the diameter of the longitudinal bar, and the stirrup spacing, respectively.

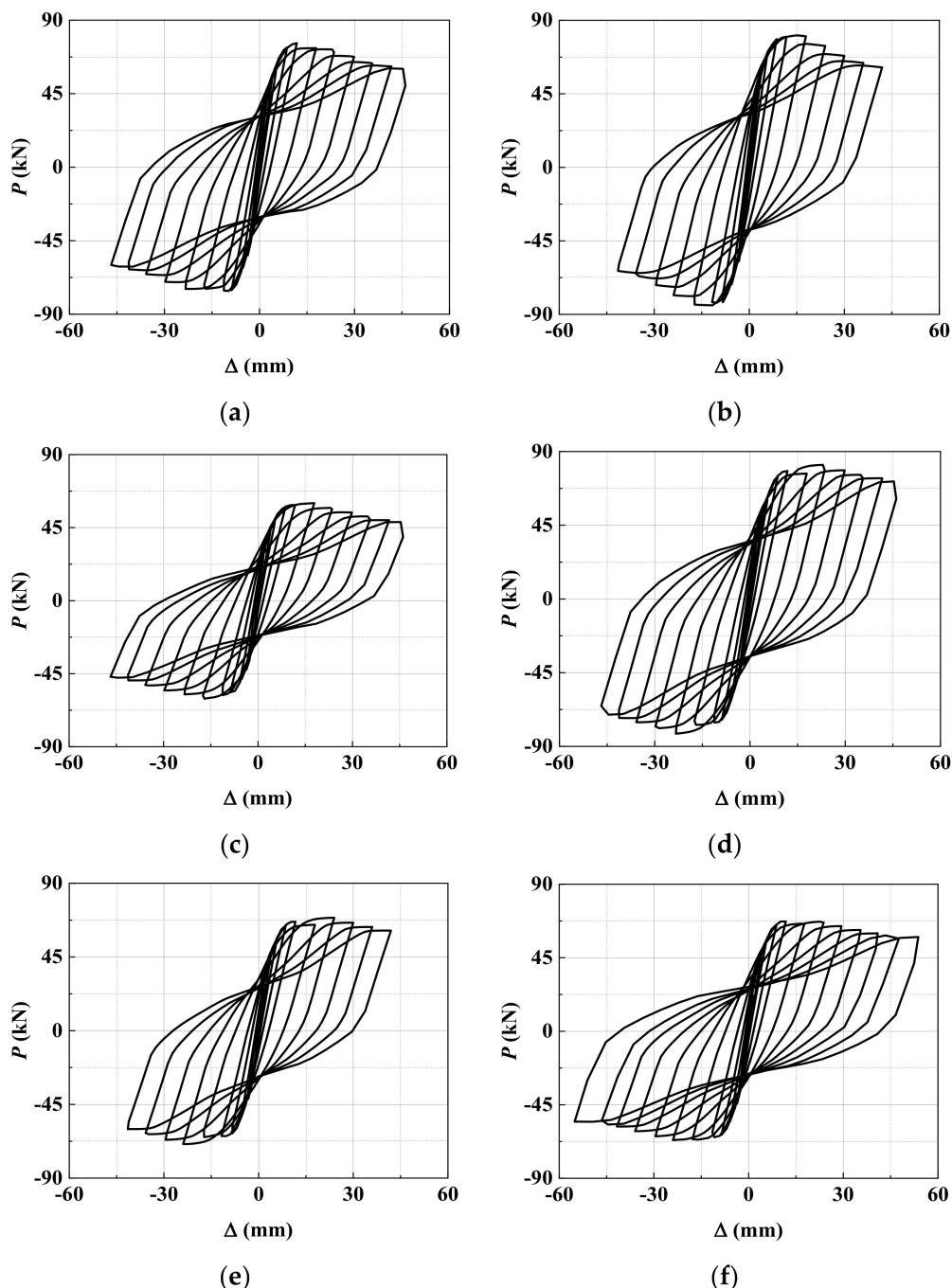

**Figure 30.** Parameter analysis by numerical simulation. (**a**) JP-A0.225. (**b**) JP-A0.3. (**c**) JP-L12. (**d**) JP-L16. (**e**) JP-J100. (**f**) JP-J50.

### 5.2.1. Skeleton Curves

The skeleton curves of the hysteretic curves were extracted as shown in Figure 31, and the characteristic values and the ductility coefficient *u* of each parameter are shown in Table 6. The ductility coefficient *u* could be defined as the ratio of the ultimate displacement to the yield displacement, i.e., $u = \Delta u / \Delta y$. The skeleton curves of the low cyclic loading test after seawater freeze–thaw damage could be summarized as follows:

(1) It was found that with the increase in the number of seawater freeze–thaw cycles, the peak load gradually decreased, but the peak displacement gradually increased, as shown in Figure 31a. When the number of seawater freeze–thaw cycles reached 125, the peak load of the skeleton curve decreased by 11%, and the peak displacement in-

creased by 40%, compared with the specimen without freeze–thaw damage. When comparing the ductility coefficients of the specimens with different numbers of seawater freeze–thaw cycles, there was no obvious relationship between the damage caused by seawater freeze–thaw cycles and the ductility coefficient, as shown in Table 6. This may be because with the increase in the number of freeze–thaw cycles, the initial stiffness and the yield load of the specimens decrease due to the increase in the yield displacement, and the increase in the peak displacement would affect the ultimate displacement at the same time due to the yield displacement and ultimate displacement in the same change trend, leading to no obvious change in the test displacement ductility coefficient.

**Table 6.** The characteristic values and the ductility coefficient.

| No. | $P_y$ (kN) | $\Delta_y$ (mm) | $P_m$ (kN) | $\Delta_m$ (mm) | $P_u$ (kN) | $\Delta u$ (mm) | $u$ |
|---|---|---|---|---|---|---|---|
| JP-F0 | 53.0 | 8.6 | 73.4 | 17.0 | 60.0 | 42.2 | 5.2 |
| JP-F1 | 51.9 | 8.7 | 74.1 | 17.8 | 58.6 | 40.2 | 5.1 |
| JP-F3 | 50.9 | 7.5 | 70.2 | 23.3 | 57.7 | 38.2 | 4.9 |
| JP-F5 | 48.1 | 8.0 | 65.8 | 26.8 | 54.5 | 40.2 | 5.0 |
| JP-A0.225 | 52.5 | 6.5 | 71.8 | 21.3 | 59.5 | 40.32 | 5.6 |
| JP-A0.3 | 56.5 | 6.8 | 78.0 | 19.8 | 63.3 | 36.1 | 4.3 |
| JP-L12 | 41.9 | 6.5 | 57.0 | 19.3 | 47.5 | 40.3 | 6.0 |
| JP-L16 | 61.0 | 8.1 | 77.8 | 26.3 | 69.1 | 40.3 | 5.5 |
| JP-J100 | 48.9 | 8.6 | 67.7 | 21.3 | 55.5 | 42.2 | 4.4 |
| JP-J50 | 54.7 | 8.7 | 67.4 | 19.8 | 59.7 | 40.2 | 5.5 |

(2) Under the same 125 seawater freeze–thaw cycles, the influences of different axial compression ratios, longitudinal reinforcement diameters, and stirrup spacings on the skeleton curve of specimens were compared, as shown in Figure 31b–d. The axial compression ratio of the low cyclic loading test increased from 0.15 to 0.3, and the shape of the skeleton curve changed from relatively "flat" to "steep"; that is, the initial stiffness of the skeleton curve increased, as shown in Figure 31b. The peak load increased by 15%, while the peak displacement decreased by 40%. As the longitudinal reinforcement diameter of specimens increased from 12 mm to 16 mm, the shape of the skeleton curve remained unchanged, as shown in Figure 31c, and the peak load and displacement increased by 27% and 20%, respectively. As the stirrup spacing of specimens decreased from 100 mm to 50 mm, the peak load remained unchanged, as shown in Figure 31d, and the ductility coefficient of the specimens increased by 20%.

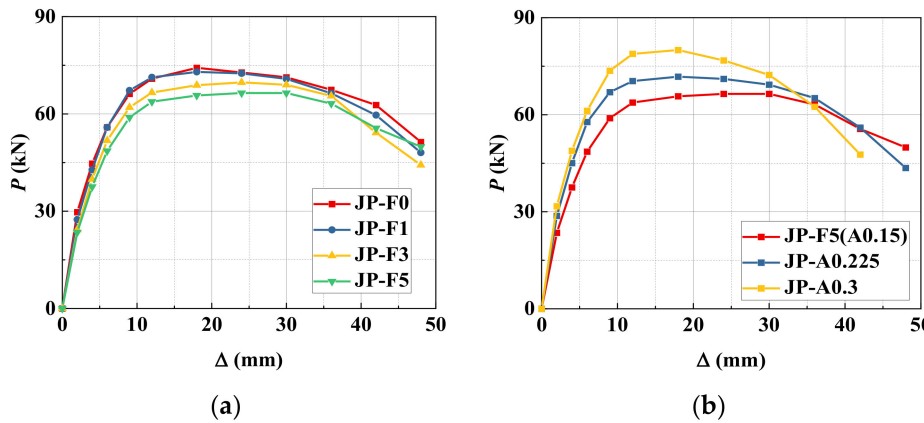

(a)　　　　　　　　　　(b)

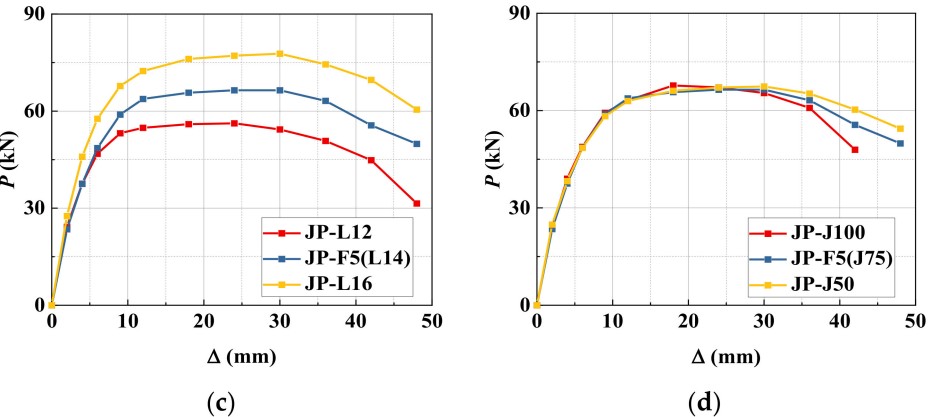

**Figure 31.** Skeleton curves. (**a**) The number of freeze–thaw cycles. (**b**) The ratio of axial compression. (**c**) Diameter of longitudinal reinforcement. (**d**) Stirrup spacing.

### 5.2.2. Stiffness Degeneration

According to the hysteresis curve of specimens after the seawater freeze–thaw cycles determined by numerical simulation, the relationship between the stiffness degradation and the loading displacement was extracted, as shown in Figure 32. The average stiffness $K_i$ of the $i$th period could be calculated by the slope of the line of load points in the push and pull direction for each load displacement amplitude, as shown in Equation (19):

$$K_i = (|P_i^+| + |P_i^-|)/(|\Delta_i^+| + |\Delta_i^-|) \tag{19}$$

where $K_i$ is the average stiffness of the ith loading cycle; $P_i^+$ and $P_i^-$ are the values of loads in the push and pull directions of the ith loading cycle, respectively; $\Delta_i^+$ and $\Delta_i^-$ are the displacements of the ith loading cycle in the push and pull directions, respectively.

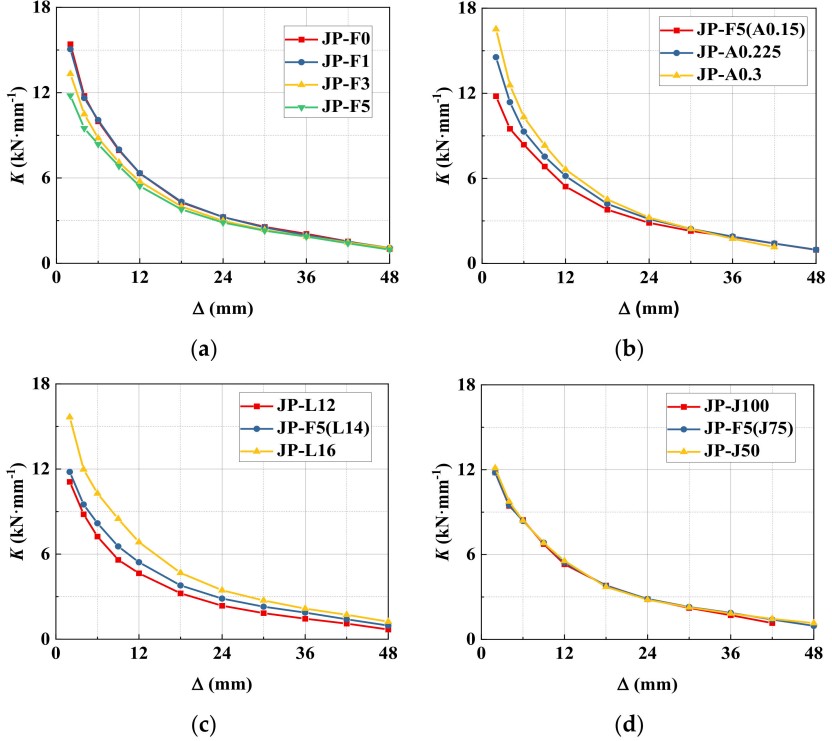

**Figure 32.** Stiffness degeneration. (**a**) The number of freeze–thaw cycles. (**b**) The ratio of axial compression. (**c**) Diameter of longitudinal reinforcement. (**d**) Stirrup spacing.

By comparing the curves of stiffness degradation under different parameters, the stiffness degradation law of specimens in the low cyclic loading test after the seawater freeze–thaw cycle could be summarized as follows:

(1) It could be seen that with the increase in the number of seawater freeze–thaw cycles, the initial stiffness of specimens gradually decreased, and the initial stiffness of specimen JP-F125 decreased by 24% compared with that of specimen JP-F0, as shown in Figure 32a. The initial stiffness of the specimens was determined by the average elastic modulus of the reinforcement and concrete of the RC pier. With the increase in the number of seawater freeze–thaw cycles, the strength and initial elastic modulus of concrete gradually decrease [29]; hence, the initial stiffness of the RC pier decreases without any change in the elastic modulus of reinforcement. When the loading displacement reached ±36 mm, the cover concrete spalled off, and the stiffness of the specimens was mainly provided by reinforcement and core concrete in the low cyclic loading test. The stiffness degradation curves of specimens with different seawater freeze–thaw cycles were the same thereafter.

(2) Under the same 125 seawater freeze–thaw cycles, the stiffness degradation curves of specimens with different axial compression ratios, longitudinal reinforcement diameters, and stirrup spacings were compared, as shown in Figure 32b–d. When the axial compression ratio increased from 0.15 to 0.3 during the test, the initial stiffness of the specimens increased by 29%, as shown in Figure 32b. When the loading displacement amplitude of the test exceeds ±30 mm, the stiffness degradation curves of the test parts with different coaxial pressure ratios coincide. As the longitudinal reinforcement diameter of the specimen increased from 12 mm to 16 mm, the initial stiffness of the specimen increased, while the trend of the stiffness degradation curve was the same, and the initial stiffness of the specimen increased by 30%, as shown in Figure 32c. Changing the stirrup spacing of the specimens had no obvious effect on the stiffness degradation of the piers, as shown in Figure 32d.

### 5.2.3. Cumulative Energy Dissipation

The energy dissipation capacity of the RC pier is evaluated by the cumulative energy dissipation in the process of low cyclic reciprocating load in this paper. According to hysteresis curves of specimens with different parameters, the relationship between the cumulative energy dissipation and the loading displacement was extracted, as shown in Figure 33. The cumulative energy dissipation value of the $i$th loading cycle was the sum of areas in three-loop hysteresis and the cumulative energy dissipation value of the $(i-1)$th loading cycle, which could be calculated by Equation (20):

$$E_p = \sum_{i=1}^{n} E_{pi} \tag{20}$$

where $E_p$ is the cumulative energy dissipation value of the RC pier; $E_{pi}$ is the cumulative energy consumption of the $i$th loading cycle; $n$ is the total loading period before the RC pier reaches the ultimate load.

By comparing the cumulative energy dissipation curves of specimens under different parameters, the energy dissipation capacity of specimens under a low cyclic loading test after the seawater freeze–thaw cycle could be summarized as follows:

(1) It could be seen that the energy dissipation capacity of specimens decreases with the increase in the number of seawater freeze–thaw cycles, as shown in Figure 33a. The total cumulative energy dissipation value of specimen JP-F125 was only 5% lower than that of specimen JP-F0, indicating that the seawater freeze–thaw cycle has little effect on the energy dissipation capacity of the RC pier.

(2) Under the same 125 seawater freeze–thaw cycles, the energy dissipation capacity of specimens with different axial compression ratios, longitudinal reinforcement diameters, and stirrup spacings were compared, as shown in Figure 33b–d. When the axial compression ratio increased from 0.15 to 0.3, the total cumulative energy dissipation value of the specimen first increased by 9% and then decreased by 16%, as shown in Figure 33b. The

main reason is that when the axial compression ratio was 0.3, the ultimate displacement of the specimen in the low cyclic loading test was lower than that when the axial compression ratio was 0.15 and 0.225. Therefore, increasing the axial compression ratio of the test would reduce the energy dissipation capacity of the specimen. When the longitudinal reinforcement diameter of specimens increased from 12 mm to 16 mm, the energy dissipation capacity of the RC pier increased, and the total cumulative energy dissipation value of the specimen increased by 30%, as shown in Figure 33c. When the stirrup spacing of the specimen decreased from 100 mm to 50 mm, the energy dissipation capacity of the specimen increased, and the total cumulative energy dissipation value of the specimen increased by 32%, as shown in Figure 33d. The main reason is that when the stirrup spacing was 100, the ultimate displacement of the specimen in the low cyclic loading test was lower than that when the stirrup spacing is 75 mm and 50 mm, so reducing the stirrup spacing would improve the energy dissipation capacity of the specimens.

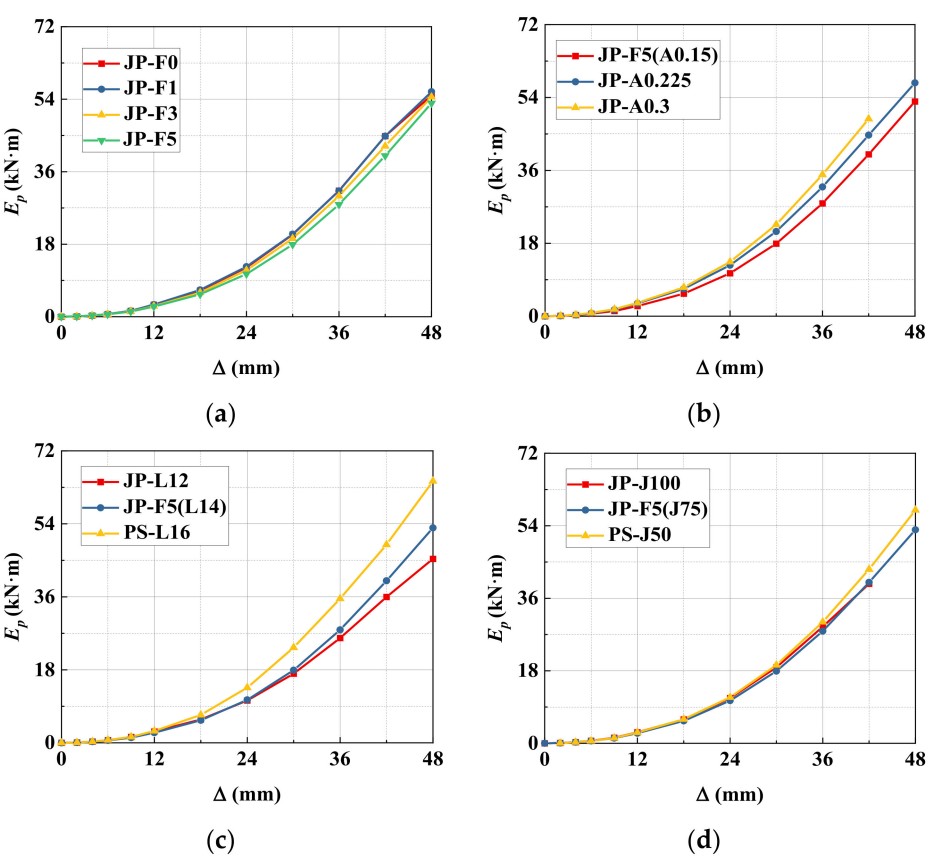

**Figure 33.** Cumulative energy dissipation. (**a**) The number of freeze–thaw cycles. (**b**) The ratio of axial compression. (**c**) Diameter of longitudinal reinforcement. (**d**) Stirrup spacing.

## 6. Conclusions

A low cyclic loading test was carried out on 12 precast segmental RC piers with different numbers of seawater freeze–thaw cycles. The constitutive relation and compression damage of concrete after seawater freeze–thaw cycles were established, and the method of meso-element equivalent and layered modeling was used in the numerical simulation of the low cyclic loading test on RC piers with freeze–thaw damage. The analysis of the experimental and numerical simulation results led to the following conclusions:

(1) The degradation law of the reloading modulus under different freeze–thaw cycles could be described by the degradation curve of non-freeze–thaw concrete, which proves the generality of the conclusion in each group of tests. The constitutive relationship of

concrete under different seawater freeze–thaw cycles could provide a theoretical basis for the heterogeneity of damage to concrete.

(2) With the increase in the number of seawater freeze–thaw cycles, the peak load decreased by 11%, and the peak displacement decreased by 40%. The initial stiffness of the specimen decreased by 24%. The total cumulative energy dissipation value of the specimen was reduced by only 5%. The ductility coefficient is not affected by the number of freeze–thaw cycles. The seismic performance of RC piers was reduced by seawater freeze–thaw cycles.

(3) Under the same number of seawater freeze–thaw cycles, i.e., 125 cycles, the peak load increased by 15%, while the peak displacement decreased by 40% with the increase in axial compression ratio. The peak load and displacement increased by 27% and 20% as the longitudinal reinforcement diameter of specimens increased. The ductility coefficient of the specimens increased by 20% when the stirrup spacing of specimens was reduced.

(4) Under the same number of seawater freeze–thaw cycles, i.e., 125 cycles, the initial stiffness of the specimens increased by 29% with the increase in axial compression ratio. The initial stiffness of the specimen increased by 30% as the longitudinal reinforcement diameter of specimens increased. The total cumulative energy dissipation value of the specimen increased by 32% when the stirrup spacing of specimens was reduced. Compared with other design parameters, increasing the diameter of longitudinal reinforcement can effectively improve the seismic performance of the pier.

## 7. Further Work

Research on the mechanical properties of RC bridges under the coupling effect of concrete freeze–thaw cycles and reinforcement corrosion will be the focus of future work. At the same time, it is necessary to design a test scheme closer to the actual situation according to the actual freeze–thaw environment of seawater.

**Author Contributions:** Conceptualization, F.T. and X.W.; methodology, W.Y.; experiment, Y.Z. and Y.L; validation, F.T.; formal analysis, F.T. and X.W.; investigation, Y.Z. and Y.L.; resources, W.Y.; data curation, F.T., X.W. and Y.Z.; writing—review and editing, F.T. and X.W.; supervision, W.Y. All authors have read and agreed to the published version of the manuscript.

**Funding:** This work was financially supported by the Natural Science Foundation Project of Liaoning Province (2022-BS-191).

**Institutional Review Board Statement:** Not applicable.

**Informed Consent Statement:** Not applicable.

**Data Availability Statement:** Not applicable.

**Acknowledgments:** This research was performed at Shenyang Jianzhu University.

**Conflicts of Interest:** The authors declare no conflict of interest.

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
