# Peer review of "Numerical Simulation of Reinforced Concrete Piers after Seawater Freeze–Thaw Cycles"

_coatings, doi:10.3390/coatings12121825_

Round 1

Reviewer 1 Report

Please consider the following comments while preparing the revised manuscript

1. Abstract Line 1-3: Sentence requires revising. Try to make simple sentences and avoid making compound sentences. Same comment is applicable throughout the manuscript.

2. Line 18-19: Please provide quantification on the results comparison between the experiments and numerical analyses.

3. Line 24-25: The sentence is redundant and is to be removed from the revised submission.

4. What does it mean by meso-element equivalent? Please clarify.

5. Introduction first sentence (Line 30-31): The statement is redundant n the context and is to be removed from the revised manuscript.

6. Knowledge gap or need for the present work is to be highlighted which is very important to access the novelty.

7. Section 2.1: Equations (2) and (3) are to be better described for the benefit of readers.

8. The authors have provided more emphasis on the compression damage phenomenon of concrete. It is not clear why they have neglected the tension damage. In general, the bridge members will be subjected to combined compression and bending loads. Hence, the tensile stresses will play a major role. Hence, similar explanation has be presented for damage in tension (dt) as well.

9. Please clarify the use of Figure 10 in the context of the contribution from the paper? Is it your own model developed? If not, Figure 9 or 10 is to be removed.

10. This study has nothing to directly determine the effect of freeze-thaw cycles. Instead, this just uses the degradation of compressive strength which is not constant and varies from test to test and intensity. This cannot be a rationale way to specify the freeze-thaw effect prediction.

11. Conclusions are presented like a real conclusion. It is just a mere summary. Provide a detailed quantitative conclusions/findings observed from the study.

12. Also, add the limitations of the present work and suggestions to improve which can be done as the scope for further work.

Author Response

The authors are very thankful to the reviewer for many useful and valuable comments and suggestions, which we believe have strengthened the quality of the manuscript. We have addressed the points raised by the reviewer, and accordingly revised the manuscript (changes are highlighted in red color).

Reviewer 2 Report

The manuscript presents a numerical study using the finite element method to analyze the combined effect of sea water freeze-thaw cycles and seismic load on offshore bridge piers. The authors have to carefully consider the recommendations and answer the questions detailed in the following notes.

1- The abstract should include the basic details of the numerical study, which is the main methodology of the article. For instance, what type of numerical analysis was used? What was the software used? The abstract should also include the most important numeral conclusions and comparisons.

2- In the introduction section, lines 120-122, the authors stated that “Therefore, it is necessary to study the freeze-thaw depth of RC piers after the seawater freeze-thaw cycle before the numerical simulation of the seismic resistance of RC piers after seawater freeze-thaw cycles.” The scientific meaning behind the sentence is lost due the too many “after” and “before” used. This sentence is supposed to show the novelty of the article, while it is not clear. Hence, the novelty of the presented numerical analysis and its difference from already existing works must be clearly highlighted.

3- Section 2.2, lines 185 to 210 describe results from a plate specimen subjected to compression loading after different freeze-thaw cycles. It is not clear if the work was conducted in this study or was cited from literature. If it is an experimental part from this study, then full details about materials used for mixtures, testing setup and the other experimental details must be introduced before this section, while if it was cited from literature, then this should clearly be clarified at the beginning of the section and reference number should be added after the figure title. Leaving this section like this defines a major deficiency that must seriously be considered by the authors.

4- In line 216, the authors cited “Liu [25].”, which means that the results presented in Figures 4 and 5 and discussed in section 2.2 are totally used from this reference. Thus, the authors must clearly follow the instructions of the previous note to follow the ethical scientific citation of literature works, even if the works were previous works of the same authors.

5- It is really confusing that there is neither a reference citation used nor anything was mentioned in the figure or figure title that refers to reference [25] in Figures 4 and 5!

6- The equivalent meso-element model shown in Figure 8c includes much more colors than the three basic colors of the random aggregate model process. These colors should also be defined, either in the text or in the figure. For instance, what types of random elements the colors red, pink, dark green, light blue and light pink refer to?

7- It is better to add a sketch that describes the steps to define the elastic modulus Emo, which is detailed in lines 417-430.

8- Again, in section 4, an experimental work on reinforced concrete columns was described. However, very limited details about experimental test setup were given, while nothing about the concrete and steel materials properties and bout the concrete mixture was mentioned. As a reader I’m really confused, where I cannot understand whether this experimental work was conducted by the authors in this work, or it was cited from literature. The title and abstract refers to only a numerical study, which means that no experimental work is included in this article. On the other hand, nothing about the citation of this work from literature was mentioned! This kind of article structure confuses readers and is totally not acceptable. The authors have to detail everything about the experimental work, if it is their own work. The title and abstract should also refer to that. Otherwise, ethical citing of every detail should be adopted to clearly identify the reference of the presented work.

9- Where are the other structural results of the experimental work??? Without introducing sufficient details about the experimental work and the obtained results, the article will not be recommended for publication.

10- In section 5, a detailed flow-chart should be added to clearly define the step-wise analysis used, where the details of the ABAQUs finite element analysis and the other steps using Matlab and statistical distributions must be described.

11- There is almost nothing about the details of the finite element analysis described in section 5.1. What are the type of elements used to model concrete, steel bars, steel ties, aggregate, cement paste, sea water,…..etc. and what are their orders? What about meshing details? What are the constitutive models used for steel and for concrete in reversed tension-compression loading? How the sea water freeze-thaw was modeled in ABAQUS, and how was sea water  distinguished from tap water in the model? What are the initial conditions and what are the boundary conditions of the finite element analysis? What about the bond between steel bars and surrounding concrete, and between cement paste and aggregate, how were they modeled in ABAQUS? All these questions must clearly be answered with full details.

12- The identification numbers of parametric numerical specimens used in Table 3 and following figures must be defined in the text or using a footnote. 

Author Response

(The authors gave the same response as above.)

Author Response

(The authors gave the same response as above.)

Round 2

Reviewer 1 Report

Most of the comments were addressed. However, few comments were not addressed to a satisfactory level of the reviewer.

Reviewer 2 Report

No more comments